# A new gene set identifies senescent cells and predicts senescence-associated pathways across tissues

Dominik Saul [1,2,3] ✉, Robyn Laura Kosinsky [4], Elizabeth J. Atkinson[5],
Madison L. Doolittle[1,2], Xu Zhang [2,6], Nathan K. LeBrasseur [2,6],
Robert J. Pignolo[1,2,6], Paul D. Robbins [7], Laura J. Niedernhofer [7], Yuji Ikeno[8],
Diana Jurk [2,6], João F. Passos [2,6], LaTonya J. Hickson[9], Ailing Xue[2],
David G. Monroe [1,2], Tamara Tchkonia [2,6], James L. Kirkland [2,6],
Joshua N. Farr [1,2,6] ✉ & Sundeep Khosla [1,2,6] ✉

Although cellular senescence drives multiple age-related co-morbidities through the senescence-associated secretory phenotype, in vivo senescent cell identification remains challenging. Here, we generate a gene set (SenMayo) and validate its enrichment in bone biopsies from two aged human cohorts. We further demonstrate reductions in SenMayo in bone following genetic clearance of senescent cells in mice and in adipose tissue from humans following pharmacological senescent cell clearance. We next use SenMayo to identify senescent hematopoietic or mesenchymal cells at the single cell level from human and murine bone marrow/bone scRNA-seq data. Thus, SenMayo identifies senescent cells across tissues and species with high fidelity. Using this senescence panel, we are able to characterize senescent cells at the single cell level and identify key intercellular signaling pathways. SenMayo also represents a potentially clinically applicable panel for monitoring senescent cell burden with aging and other conditions as well as in studies of senolytic drugs.

Cellular senescence is now recognized as a fundamental mechanism of aging in animals and humans. Accumulation of DNA damage and/or other cellular stressors[1–4] causes proliferating[5,6] as well as terminally differentiated, non-dividing cells[7–10] to undergo senescence. Characteristics of senescent cells include profound chromatin and secretome changes, along with increased expression of a number of senescence markers, including *Cdkn2a/p16^{Ink4a}* and *Cdkn1a/p21^{Cip1}*, immune evasion, and resistance to apoptosis[1,11]. Senescent cells can develop a senescence-associated secretory phenotype (SASP), consisting of pro-inflammatory cytokines, chemokines, extracellular matrix-degrading proteins, and other factors that have deleterious paracrine and systemic effects[12–15]. Further, because senescent cells accumulate in multiple tissues in temporal and spatial synchrony with age-associated functional decline in both animals and humans[5,6,16], they have been hypothesized to drive the deterioration linked to numerous chronic diseases[1].

[1]Division of Endocrinology, Mayo Clinic, Rochester, MN 55905, USA. [2]Robert and Arlene Kogod Center on Aging, Mayo Clinic, Rochester, MN 55905, USA. [3]Department of Trauma, Orthopedics and Reconstructive Surgery, Georg-August-University of Goettingen, Goettingen, Germany. [4]Division of Gastroenterology and Hepatology, Mayo Clinic, Rochester, MN 55905, USA. [5]Department of Quantitative Health Sciences, Mayo Clinic, Rochester, MN, USA. [6]Department of Physiology and Biomedical Engineering, Mayo Clinic, Rochester, MN, USA. [7]Institute on the Biology of Aging and Metabolism, Department of Biochemistry, Molecular Biology and Biophysics, University of Minnesota, Minneapolis, MN, USA. [8]Department of Pathology, University of Texas Health, San Antonio, TX, USA. [9]Division of Nephrology and Hypertension, Mayo Clinic, Jacksonville, FL, USA. ✉e-mail: saul.dominik@mayo.edu; farr.joshua@mayo.edu; khosla.sundeep@mayo.edu

Importantly, the SASP as a feature of cellular senescence represents not just a locally or systemically detrimental set of factors that, in the aging organism, cause physical, metabolic, and cognitive decline[17–21], but is also a therapeutic target of interest[22–24]. Thus, given the broad availability of next-generation sequencing, there is considerable interest in monitoring responses to senolytic treatments. However, this has been challenging, especially at the single cell level[25]. In part, this is due to an imprecise definition of the heterogeneous population of senescent cells and their associated SASP which complicates appropriate monitoring of senescent cell clearance.

Due to variations in the composition of a "senescence gene set" in the current literature, in the present study we sought to identify commonly regulated genes in various age-related datasets in a transcriptome-wide approach that included whole-transcriptome as well as single cell RNA-sequencing (scRNA-seq)[26]. Based on an extensive review of the literature, we defined a panel of 125 genes as our senescence gene set ("SenMayo"), which we then validated in our own as well as publicly available datasets of tissues from aged humans and mice, including changes in this gene set following the clearance of senescent cells. Recognizing the difficulty of identifying senescent cells within scRNA-seq analyses, we next applied SenMayo to available scRNA-seq data from human and murine bone marrow/bone hematopoietic and mesenchymal cells, ascertained the identity of the senescent cells in these analyses, and characterized the communication patterns of senescent hematopoietic or mesenchymal cells with other cells in their microenvironment. Finally, we experimentally validated key predictions from our in silico analyses in a mouse model of aging and following genetic clearance of senescent cells.

## Results

### Development and validation of SenMayo in human datasets

We first analyzed previously published[27,28] as well as unpublished (see Methods) transcriptome-wide mRNA-seq analyses of human whole bone biopsies. These included bone and bone marrow (Cohort A)[27] as well as bone biopsies that were processed to remove bone marrow and bone surface cells and were thus highly enriched for osteocytes (Cohort B)[28] from young vs. elderly women (Fig. 1a). We used transcriptional regulatory relationships[29] to evaluate whether senescence- and SASP-associated pathways were enriched with aging in humans and noted enrichment of genes regulating inflammatory mediators, including *NFKB1*, *RELA*, and *STAT3* (Fig. 1b). As expected, both aged cohorts displayed an upregulation of senescence- and SASP markers such as $CDKN1A/p21^{Cip1}$, *CCL2*, and *IL6* (Fig. 1c). It should be noted that some canonical markers of senescence, including $CDKN2A/p16^{Ink4a}$, did not show the predicted increase with aging due to comparatively low expression levels. Given the limitations of single gene analyses to predict the complex mechanisms of cellular aging, we next tested whether a previously published combination of senescence/SASP genes (R-HSA-2559582) is enriched in our aging cohorts. However, this Gene Set Enrichment (GSEA)-based approach failed to predict an age-related senescence/SASP increase in either cohort (Fig. 1d).

In order to develop a more robust gene panel associated with cellular senescence, we next generated a novel gene set to predict the expression of senescence-related genes by performing an in-depth, rigorous literature search (see Methods for details of how these genes were selected). The result was a novel senescence gene set of 125 genes (SenMayo) that consisted predominantly of SASP factors (n = 83) but also included transmembrane (n = 20) and intracellular (n = 22) proteins (see Supplementary Data 1 for the complete SenMayo gene list [human and mouse]). Within this SenMayo gene set, which comprised 9 distinct clusters, cytokines/chemokines were the most densely connected regulators according to the number of descendent proteins in STRING analysis (Fig. 1e, f; network characteristics can be

found in Supplementary Data 2). Predominant connectivity (whole network density: 0.277, PPI <0.0001) was shown by *IL1A, CXCL8, CCL2* (cytokines/chemokines, blue), *IGF1* (growth factor, green), *C3* and *IGFBP4* (protease inhibitor, turquoise), *TNFRSF1A, EGF* and *EGFR* (transmembrane signal receptors, red), and *MMP2, PLAT*, and *HGF* ([metallo-]proteases, grey) (Fig. 1f). The key regulatory elements of the SenMayo genes, according to iRegulon[30], featured the Factorbook-NFKB1 motif (Supplementary Fig. 1a), and BCL3, a key transcriptional coactivator for NFKB[31] (Supplementary Fig. 1b–c), represented the leading transcription factor for a majority of SASP genes (Supplementary Fig. 1d).

Notably, when testing the enrichment of SenMayo within our two human mRNA-seq cohorts, senescence/SASP genes were significantly enriched in the bone samples obtained from elderly women (p = 0.002 [Cohort A] and p = 0.003 [Cohort B]; Fig. 1g). Using Cohort A as an example, within the R-HSA-2559582 gene set, 2 out of 50 available genes were significantly enriched in the biopsies from elderly women (Supplementary Fig. 2a), while 13 out of 120 available genes of the SenMayo gene set were significantly enriched in the elderly women (Supplementary Fig. 2b). Note that the GSEA analysis includes not only genes that differ significantly between groups, but also evaluates overall trends for differences in gene expression between groups and hence provides considerably greater power than examining individual genes[32]. The canonical SASP markers *CCL24, SEMA3F, FGF2*, and *IGFBP7* were consistently enriched in Cohort A (Supplementary Fig. 2c) and Cohort B (Supplementary Fig. 2d). In addition, *SEMA3F* was significantly correlated with the senescence marker, $CDKN1A/p21^{Cip1}$, in both cohorts (Supplementary Fig. 2e, f).

### SenMayo is applicable across tissues and species

To evaluate the applicability of SenMayo across tissues and species, we next analyzed publicly available mRNA-seq data from brain tissue isolated from young vs. aged mice (GSE145265[33], GSE128770[34], GSE94832[35], Fig. 2a–c). As is evident, aged mouse brain cells (microglia) and regions (prefrontal cortex, dorsal hippocampus) displayed a highly significant enrichment of senescence/SASP genes using the SenMayo gene list (p = 0.005, p = 0.001, p < 0.001, respectively), while the previously published gene set (R-HSA-2559582) did not reach statistical significance (p = 0.157, p = 0.117, p = 0.192, respectively). In addition, using murine bone marrow from the *tabula muris senis* (a murine single cell transcriptome atlas of young vs. aged tissues[36]), the applicability of SenMayo in identifying senescent cells associated with aging was confirmed by GSEA (Fig. 2d). Thus, SenMayo identifies senescent cells associated with aging across tissues (bone/bone marrow and brain) and species (humans and mice).

### SenMayo demonstrates clearance of senescent cells

In order to independently validate our in silico analyses, we next made use of our previously described *p16-INK-ATTAC* mouse model that allows for inducible clearance of $p16^{Ink4a}$-expressing senescent cells after administration of the drug AP20187 (AP)[37]. In previous studies, we have demonstrated increases in $Cdkn2a/p16^{Ink4a}$ and $Cdkn1a/p21^{Cip1}$ mRNA levels with aging in bones from these mice[7] as well as reductions in these mRNAs following clearance of senescent cells in *p16-INK-ATTAC* mice treated with AP and concordant changes in other markers of cellular senescence (e.g., telomeric DNA damage markers in osteocytes)[37]. Importantly, in young vs. old mice, SenMayo was expressed at a significantly higher level in bones from the old mice (Fig. 3a) and was significantly reduced following AP treatment of old *p16-INK-ATTAC* mice (Fig. 3b). Moreover, by using the SenMayo genes, a higher overlap of young vs. old + AP-treated mice as compared to young vs. old + vehicle-treated mice was observed through principal component analysis (PCA) (Fig. 3c).

We further validated the ability of SenMayo to predict senescent cell clearance by examining a human cohort. In a phase I pilot study,

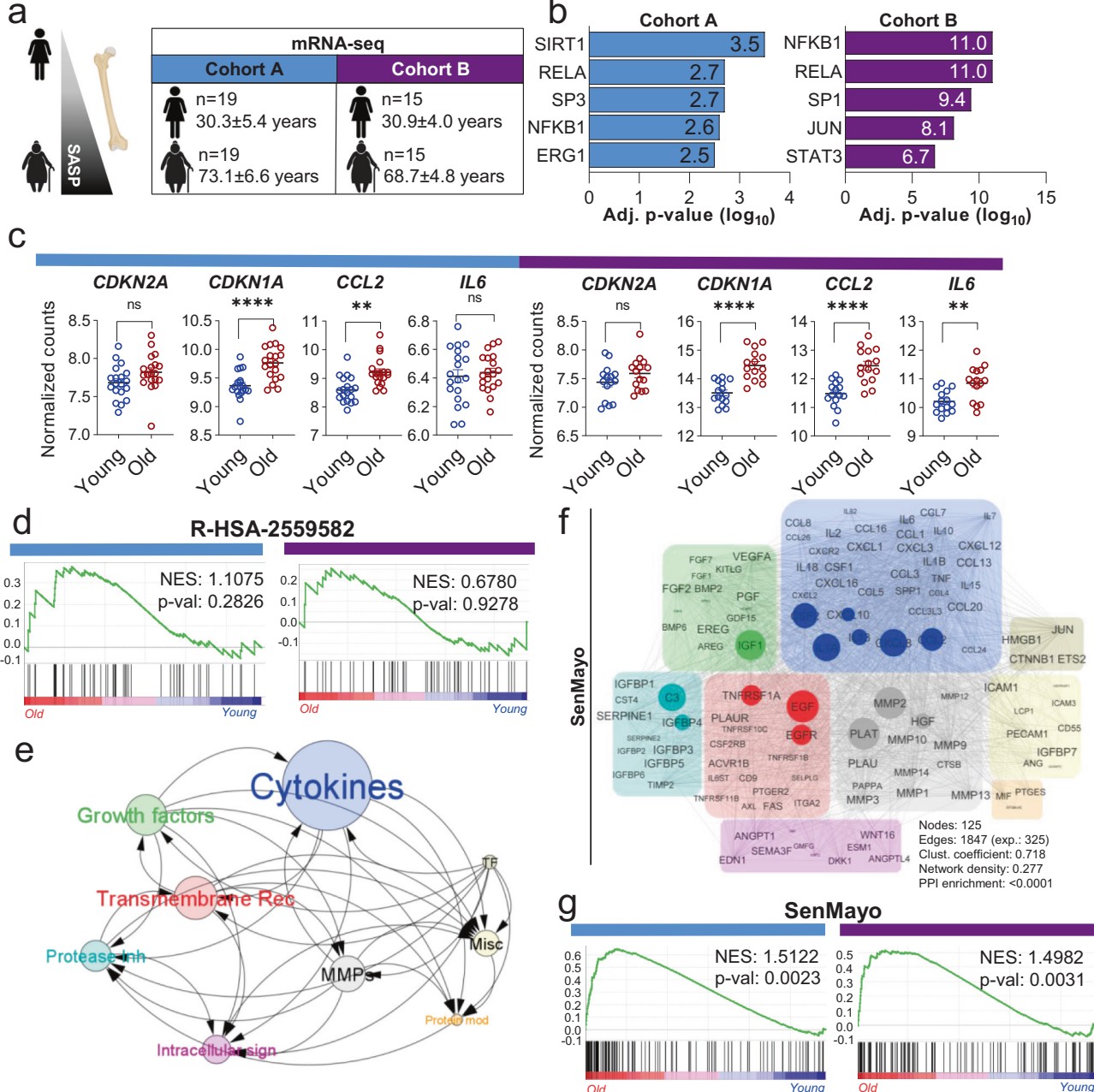

**Fig. 1 | Development and validation of the SenMayo gene set. a** Human samples from Cohort A (bone and bone marrow biopsies) and cohort B (highly enriched osteocyte fractions) were used for transcriptome-wide RNA-seq analyses; **b** Making use of TRRUST analyses[30], we found several inflammation- and stress-associated genes, including *SIRT1* and *NFKB1*, to be upregulated in the elderly women; *p* values were adjusted according to Benjamini–Hochberg. **c** In both gene sets, *CDKN1A/ P21Cip1* and several SASP markers such as *CCL2* and *IL6* showed consistent upregulation with aging, while *CDKN2A/p16Ink4a* (due to comparatively low expression) did not change significantly. Two-sided unpaired t-tests except for CCL2, where a Kolmogorov–Smirnov test was used (cohort A: *CDKN2A: p* = 0.0834; *CDKN1A: p* < 0.0001; *CCL2:* p = 0.0034; *IL6: p* = 0.6391; cohort B: *CDKN2A: p* = 0.1658, *CDKN1A: p* < 0.0001; *CCL2:* p < 0.0001, *IL6: p* = 0.0017). **d** The commonly used senescence/SASP gene set (R-HSA-2559582) failed to predict the aging process in either human cohort. Nominal *p* value, calculated as two-sided t-test, no adjustment since only one gene set was tested; **e** The SenMayo gene set includes growth factors, transmembrane receptors, and cytokines/chemokines that are highly influenced by other members of the gene set. The circle size depicts groupwise interactions, arrows point the direction of these interactions. **f** SenMayo encodes a dense network of nine different protein classes within a strong interaction network. The size of each circle represents the connectivity with other members of the gene set, grey lines represent interactions[92]; **g** Genes included in the SenMayo gene set were significantly enriched with aging in both human cohorts. Nominal *p* value, calculated as two-sided *t*-test, no adjustment since only one gene set was tested Cohort A: *n* = 38 (19 young, 19 old, all ♀), Cohort B: *n* = 30 (15 young, 15 old, all ♀). ***p* < 0.01, *****p* < 0.0001. Fig. 1a was designed using Biorender.com. Depicted are mean ± SEM. Source data are provided as a Source Data file.

the senolytic combination of Dasatinib plus Quercetin (D + Q)[38] was administered to subjects with diabetic kidney disease for three consecutive days[24,39]. Diabetic kidney disease was chosen as a model in the original trial because both obesity (associated with type 2 diabetes mellitus) and chronic kidney disease are linked to an increase in

senescent cell burden[40,41]. We performed RNA-seq from adipose tissue samples obtained from these subjects before and 11 days after D + Q treatment (male: female = 7:2, age: 68.7 [±3.1] years, Fig. 3d)[24,39]. As shown in Fig. 3e, there was a significant reduction in SenMayo (*p* = 0.002) in the subcutaneous adipose tissue samples in the subjects

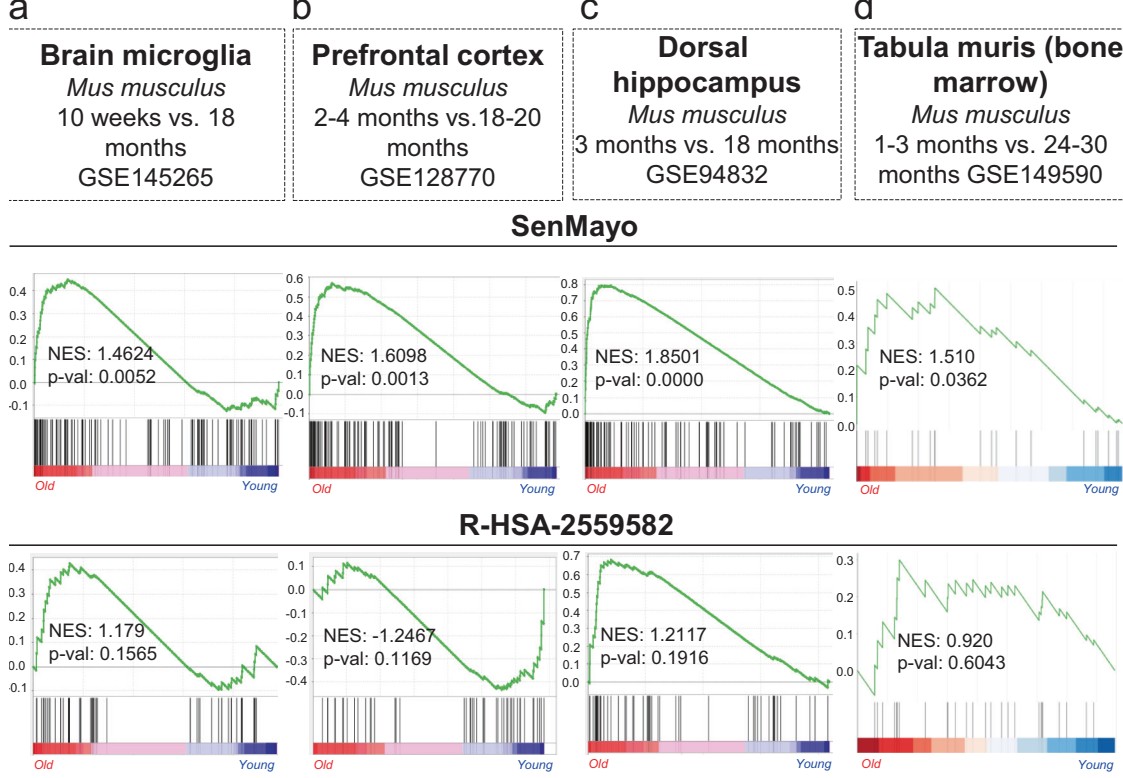

**Fig. 2 | The SenMayo gene set predicts aging across tissues and species.**
**a** Compared to the conventional gene set, the SenMayo list is significantly enriched during the aging process in murine brain microglia ($p = 0.1565$ vs. $p = 0.0052$; GSE145265), $n = 4$ (2 young, 2 aged, all ♂), **b** murine prefrontal cortex ($p = 0.1169$ vs. $p = 0.0013$; GSE128770), $n = 48$ (24 young, 24 aged, all ♂), and (**c**) murine dorsal hippocampus ($p = 0.1916$ vs. $p < 0.001$; GSE94832), $n = 12$ (6 young (3 ♀), 6 aged (2♀). Likewise, the murine bone marrow (**d**) within the *tabula muris senis* (GSE149590[102]) has a higher enrichment of the SenMayo genes within the old cohort ($p = 0.6043$ vs. $p = 0.0362$), $n = 11$ (4 young (2 ♂, 2 ♀), 7 old (7 ♂, 0 ♀). Nominal $p$ values were calculated as two-sided $t$-test, no adjustment since only one gene set was tested. Source data are provided as a Source Data file.

following D + Q treatment, consistent with a reduction in senescent cell burden, which was independently validated by demonstrating reductions in p16[Ink4]+, p21[Cip]+, and SA-βgal+ cells in the adipose tissue biopsy samples following D + Q treatment[24,39]. Thus, these direct interventional studies in mice and humans demonstrate that not only is SenMayo associated with aging, but it is also reduced following clearance of senescent cells.

## SenMayo outperforms existing senescence/SASP gene sets
In addition to directly comparing SenMayo to the R-HSA-2559582 senescence/SASP gene set, we also compared it to five additional senescence/SASP gene sets[42–46] in all of the mouse and human models described above. As shown in Table 1, SenMayo consistently outperformed these gene sets (based on normalized enrichment scores [NES] and $p$ values) both in the ability to identify senescent cells with aging across tissues and species and in demonstrating responses to senescent cell clearance.

## SenMayo identifies senescent cells in scRNA-seq datasets
Although scRNA-Seq provides important information regarding changes in gene expression at the individual cell level, it has been problematic for evaluating cellular senescence in a given cell. In part, this is because the *Cdkn2a/p16[Ink4a]* mRNA is expressed at relatively low levels, even in senescent cells[47], and may not be reliably detected in scRNA-seq data. Although *Cdkn1a/p21[Cip1]* is generally expressed at higher levels in RNA-seq data, presence or absence of *Cdkn1a/p21[Cip1]* also may not consistently identify a senescent cell[44]. As such, having validated SenMayo as being associated with cellular senescence in the context of aging, we next tested whether it could identify senescent

cells at the single cell level. To evaluate this first for hematopoietic cells, we analyzed publicly available single cell bone marrow datasets from 20 healthy donors across a broad age range (24–84 years)[48] and evaluated 68,478 hematopoietic cells for expression of the SenMayo gene set (GSE120446)[48], Fig. 4a (the key genes for clustering are demonstrated in Supplementary Fig. 3a).

This analysis detected multiple cellular clusters that were more highly enriched than others for senescence/SASP genes, *i.e.*, had higher enrichment scores (ES). These high ES clusters included CD14[+] and CD16[+] monocytes as well as macrophages (Fig. 4a, Supplementary Fig. 3b). By selecting the top 10% of cells with the highest expression of senescence/SASP-associated genes, we generated a cluster of cells, consisting of 6,850 cells, predominantly of monocytic origin (referred to as "SASP cells" in Fig. 4b). These SASP cells showed an increase in canonical markers of senescence such as *CDKN1A/p21[Cip1]* and *TGFB1*, which are independent and not included in the SenMayo gene set, as well as enrichment of previously published gene sets indicative of human[49] and cellular aging[50] (Table 2). Visually, the SASP cells had a high correlation with genes in two established aging gene sets (GenAge and positively regulated in CellAge, Fig. 4b; the geneset-to-geneset comparison is shown in Supplementary Fig. 3c, d). To further elucidate the replicative state of these cells, we compared their cell cycle state with the other clusters. A shift towards the G1 phase occurred within the SASP cells (Supplementary Fig. 3e), consistent with replicative arrest. This finding was supported by cell cycle arrest gene enrichment within the SASP cells (Supplementary Fig. 3f). In addition, pseudotime analysis (Supplementary Fig. 3g, left panel), which permits elucidation of the temporal gene expression pattern of a specific cell type, revealed an increase in SASP cells over time (representing

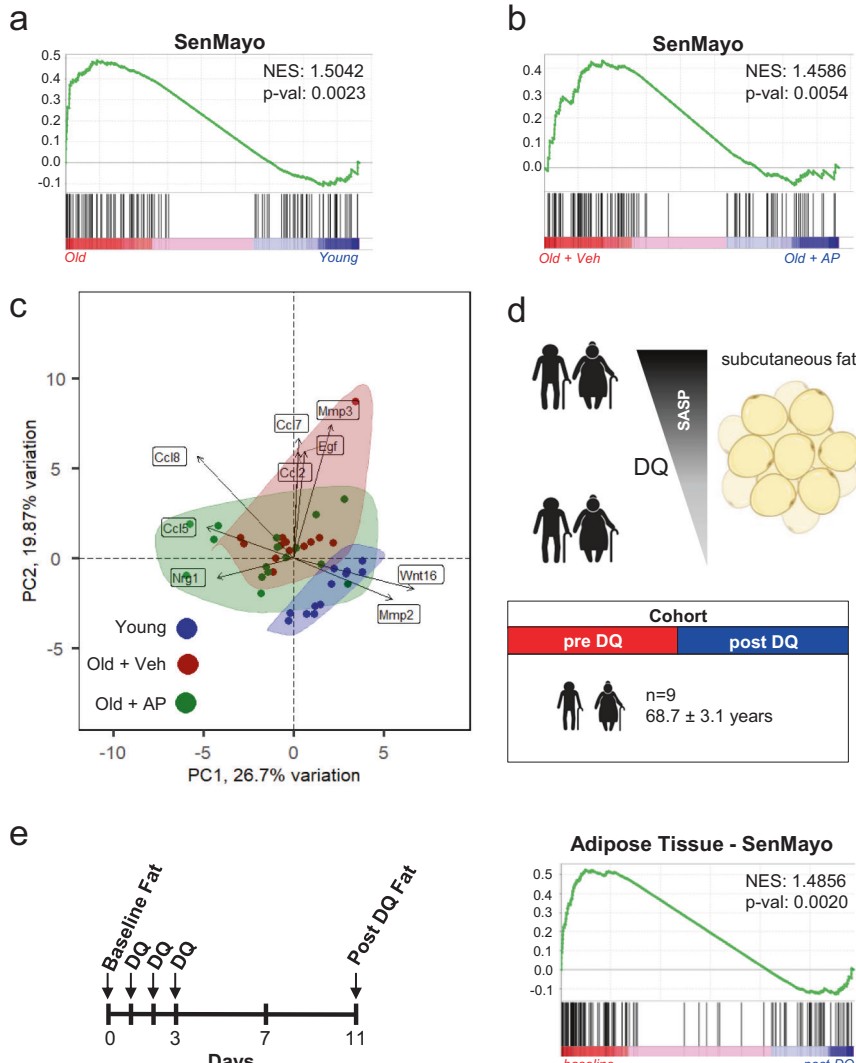

**Fig. 3 | The SenMayo gene set tracks genetic and pharmacologic clearance of senescent cells. a** The SenMayo panel successfully indicated aging in bone in mice (*p* value = 0.0023), *n* = 25 (12 young, 13 old (all ♀). Nominal *p* value, calculated as two-sided *t*-test, no adjustment since only one gene set was tested; **b** The elimination of *p16*^Ink4a-expressing senescent cells by AP20187 administration was shown previously to reverse the aging bone phenotype[37]. The SenMayo gene set successfully demonstrated the significant reversal of the aging phenotype at the gene expression level upon the elimination of *p16*^Ink4a-expressing senescent cells (*p* = 0.0054), *n* = 29 (13 Veh, 16 AP (all ♀). Nominal *p* value, calculated as two-sided *t*-test, no adjustment since only one gene set was tested; **c** By specifically using the expression patterns of the SenMayo gene set, our bone RNA-seq revealed no similarities in gene expression patterns between young (blue) and old + veh (red)

treated mice, and a substantial overlap of expression profiles of old + AP (green) mice with young mice. The highlighted genes represent variables, and the arrows drawn from the origin indicate their "weight" in different directions, according to the theories of Gabriel[103]; **d** We used a previously published mRNA-seq dataset from human adipose tissue of our group[24, 39], to evaluate changes in SenMayo following D + Q treatment. Adipose tissue was collected before and 11 days after three days of oral D + Q treatment. Figure was designed using Biorender.com; **e** Using SenMayo, there was a reduction of SenMayo (*p* = 0.002) in the subcutaneous fat samples in the subjects treated with D + Q, consistent with a reduction in senescent cell burden following D + Q treatment (*n* = 9 (7 ♂, 2 ♀)). Nominal *p* value, calculated as two-sided *t*-test, no adjustment since only one gene set was tested. Source data are provided as a Source Data file.

---

differentiation), particularly in CD14⁺ monocytes, CD16⁺ monocytes, and macrophages (Supplementary Fig. 3g, middle panel).

In addition to intracellular signaling pathways differentially regulated in SASP-secreting cells, these cells have been demonstrated to affect surrounding cells[13,51]. To explore these intercellular interactions, we evaluated potential ligand-receptor interactions and secretion patterns based on underlying gene expression levels in different hematopoietic cell types in human bone marrow[52]. The strongest interaction of SASP cells was found with T cells, followed by monocytic cells and B cells (Fig. 4c). Among the affected pathways, the major histocompatibility complex class I (MHC-I), Macrophage Migration Inhibitory Factor (MIF), and Platelet And Endothelial Cell Adhesion Molecule 1 (PECAM1, CD31) pathways were most highly enriched (Fig. 4d, e, Supplementary Fig. 4a, b). Of note, in the pseudotime

analysis described above, *MIF* expression also increased markedly in terminally differentiated CD14⁺ and CD16⁺ monocytes and macrophages and SASP cells (Supplementary Fig. 3g, right panel). Moreover, MIF pathway members including *CD74*, *CXCR4*, and *CD44* had overall high expression in SASP cells (Fig. 4e). Compared to other cell types, the overall outgoing interaction strength of SASP cells was remarkably high (Supplementary Fig. 5a). Besides their importance as senders, mediators, and influencers (defined by signalling network analysis using centrality measures; for details see[52,53], Supplementary Fig. 5b, c), SASP cells displayed a substantial incoming signaling pattern dominated by the MIF, ANNEXIN, CD45, IGGB2, MHC-I, MHC-II, and PECAM1 pathways (Supplementary Fig. 5d). Within these SASP cells, the strongest direct receptor-ligand MIF interaction between the ligand CD74 and the receptor CD44 was mainly detected in other monocytic

**Table 1 | Comparison of SenMayo with six existing senescence/SASP gene sets**

| | Human aging | | | | Mouse aging | | | | | | Bone marrow | | Mouse genetic clearance of senescent cells | | | | Human pharmacological clearance of senescent cells | |
| | Cohort A | | Cohort B | | Microglia | | Prefrontal cortex | | Dorsal hippocampus | | | | Mouse INK-ATTAC (old vs young) | | Mouse INK-ATTAC (old, vehicle vs AP) | | Adipose (Control vs D + Q) | |
| | NES | p value | NES | p value | NES | p value | NES | p value | NES | p value | NES | p value | NES | p value | NES | p value | NES | p value |
|---|---|---|---|---|---|---|---|---|---|---|---|---|---|---|---|---|---|---|
| R-HSA-2559582 | 1.1075 | 0.2826 | 0.6780 | 0.9278 | 1.1794 | 0.1565 | −1.2467 | 0.1169 | 1.2117 | 0.1916 | 0.9200 | 0.6043 | 1.4235 | 0.0326 | 1.0006 | 0.4442 | 1.1176 | 0.2356 |
| Casella_up | 1.0089 | 0.4748 | 0.6970 | 0.8885 | 0.9737 | 0.4790 | 0.9537 | 0.5209 | 1.3949 | 0.0593 | 0.8874 | 0.6339 | −1.0861 | 0.3168 | 0.6865 | 0.9627 | −0.7959 | 0.8344 |
| Purcell | 0.9329 | 0.5944 | 0.9585 | 0.5093 | 1.5120 | 0.0178 | 1.1731 | 0.2304 | 1.8117 | 0.0000 | 1.5224 | 0.0419 | 1.3894 | 0.0778 | −0.7300 | 0.8874 | 0.9132 | 0.5852 |
| Hernandez | 0.7849 | 0.7771 | 0.7846 | 0.7802 | 0.7146 | 0.9461 | 0.8662 | 0.6710 | −0.8100 | 0.7650 | −0.5778 | 0.9789 | 1.2513 | 0.1718 | 1.4796 | 0.0266 | 0.9476 | 0.5266 |
| Fridman_up | 1.4249 | 0.0169 | 1.5407 | 0.0174 | 1.4397 | 0.0206 | 0.9639 | 0.5449 | 1.7482 | 0.0000 | 1.6100 | 0.0163 | 1.0762 | 0.3145 | 1.3413 | 0.0347 | 1.4187 | 0.0122 |
| Sencan | −0.8460 | 0.9362 | 0.8038 | 0.8235 | 0.8674 | 0.8144 | 1.0328 | 0.4006 | 1.5302 | 0.0011 | 0.7247 | 0.8667 | 0.8838 | 0.7312 | −1.0674 | 0.3182 | 0.8643 | 0.8983 |
| SenMayo | 1.5122 | 0.0023 | 1.4982 | 0.0023 | 1.4624 | 0.0052 | 1.6098 | 0.0013 | 1.8501 | 0.0000 | 1.5100 | 0.0362 | 1.5042 | 0.0023 | 1.4586 | 0.0054 | 1.4856 | 0.0020 |

Note that in GSEA analyses, p values < 0.25 are considered potentially significant, although we also identified p values <0.05 and <0.01 (NES, normalized Enrichment Score). Nominal p value, calculated as two-sided t-test, no adjustment since only one gene set was tested.

cells, while the MIF interaction via the ligand CD74 and receptor CXCR4 pair was significant for SASP to CD10$^+$ B and CD20$^+$ B cells as well as plasmacytoid dendric cells. The PECAM1 pathway targeted plasma cells and CD16$^+$ monocytes (Supplementary Fig. 5e).

Further analysis revealed that the SASP cells were characterized by distinct patterns of co-expression out of which several markers were found to be strongly associated with each other (Fig. 4f–g)––e.g., *EREG/IL1B, ICAM1/CDKN1A*, and *JUN/CDKN2A*. Out of the 125 genes within the SenMayo panel, some were consistently upregulated (red in Fig. 4f), while others were simultaneously downregulated (blue in Fig. 4f). After we found that some of the "canonical" SASP markers such as *EREG/IL1B* and SASP/senescence markers such as *ICAM1/CDKN1A* showed high concordance in their cell-wise expression patterns, we aimed to find surrogate genes for certain low-expressed genes––e.g. *CDKN2A/p16^{Ink4a}*. Within the SASP cluster, we found a strong correlation between *JUN* and *CDKN2A/p16^{Ink4a}* expression, which represents a potential approach to overcome the challenge of low *CDKN2A/p16^{Ink4a}* expression in sequencing datasets. To independently confirm these correlations, we depicted these genes in a pairwise fashion with kernel density estimation within the SASP cell clusters (Fig. 4h), where the overall SASP cells are in blue and the red/yellow colors indicate higher levels of expression within the SASP cells of each gene[47]. These analyses thus demonstrate the validity of the SenMayo gene set in a human bone marrow scRNA-seq dataset and identify monocytic cells as the hematopoietic cell population with the highest proportion of SASP-associated cells.

To further test SenMayo in single cell datasets and potentially contrast bone marrow hematopoietic cells to bone/bone marrow mesenchymal cells, we next evaluated a published murine dataset that contained scRNA-seq data from bone and bone marrow mesenchymal cells (GSE128423[54], Fig. 5a, *n* = 35,368 cells; the key genes for clustering are demonstrated in Supplementary Fig. 6a). We detected a heterogenous distribution of highly enriched cells for SenMayo ("SASP cells", *n* = 3537), which likewise were enriched in both GenAge and CellAge (Fig. 5b, Supplementary Fig. 6b, c), canonical markers of senescence (*Cdkn1a/p21^{Cip1}* and *Tgfb1*, Fig. 5b) and was primarily comprised of cells from the osteolineage (OLC1 and 2) as well as leptin receptor-positive (Lepr$^+$) MSC cluster (Supplementary Fig. 7a shows the fraction of the original clusters that were subsequently assigned to the newly created SASP cluster and Supplementary Fig. 4b indicates the percentage of cells within each cluster that were in the top 10% of cells enriched for SenMayo genes). Interestingly, 21% of osteolineage cells (24% in OLC 1 and 18% in OLC2) had the highest enrichment for SASP factors (Supplementary Fig. 7b). Similar to the human hematopoietic bone marrow dataset, murine bone/bone marrow mesenchymal SASP cells displayed a shift in cell cycle phase to the G1 phase (Supplementary Fig. 7c). This was confirmed by gene ontology analysis revealing enrichment of senescence- and cell cycle arrest-associated genes in these cell clusters (Supplementary Fig. 7d). The murine mesenchymal SASP cells were characterized by a high interaction with osteolineage and chondrocytic cells (Fig. 5c), with the MIF and PECAM1 pathways again among those significantly enriched, where these cells mostly acted as senders and influencers (Fig. 5d, Supplementary Fig. 7e). Notably, SASP cells had one of the highest outgoing interaction strengths (Supplementary Fig. 7f). A direct communication of these mesenchymal SASP cells mostly appeared in the MIF pathway (via L/R Mif/Ackr3) with chondrocytic cells and mineralizing osteocytes (Supplementary Fig. 7g). Interestingly, and as predicted from the human RNA-seq data (Fig. 1b, Supplementary Fig. 1), a major regulator of the SASP cells was the transcription factor BCL3, a key transcriptional coactivator for NFkB[31] (Supplementary Fig. 8a, b).

The three main origins for the SASP cluster (namely Lepr$^+$ MSCs, OLC 1, and OLC 2), as depicted in pseudotime, demonstrated that the SASP cells accumulated in a terminal developmental branch, coinciding with increased *Cdkn1a/p21^{Cip1}* and *Trp53*

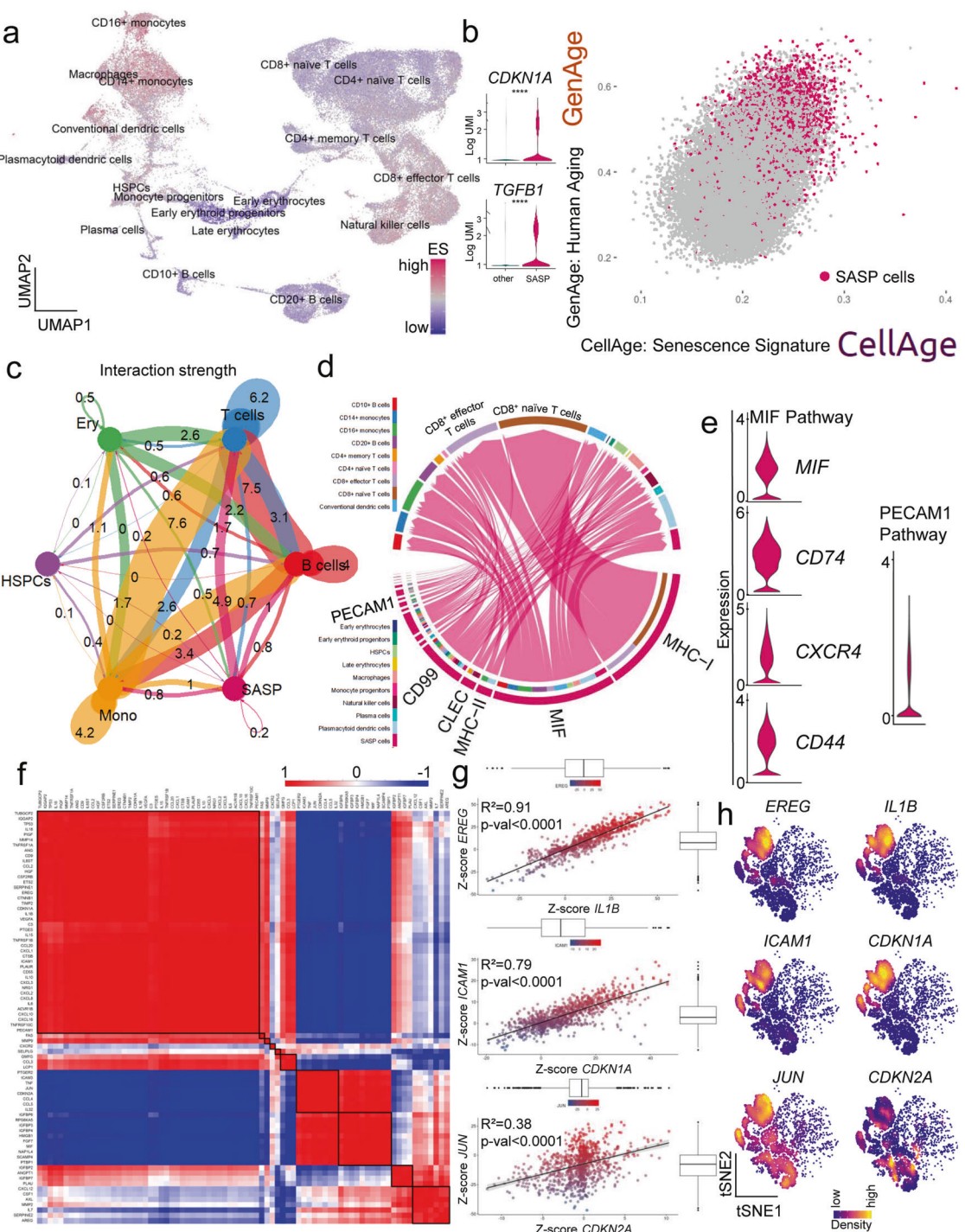

expression (Fig. 5e, Supplementary Fig. 9a, b). Further analysis of these pseudotime expression patterns showed that certain genes followed defined modules (green, blue, and red in Fig. 5f), which then formed co-expressional patterns (Supplementary Fig. 10a). Within the SASP cluster, these co-expressional patterns could be imaged at an individual cell level, predicting genes of similar abundance within some cells (Fig. 5g). For example, while *Pappa* and *Fgf7* were simultaneously downregulated in terminally differentiated stages (Fig. 5f, blue color in the green cluster, Fig. 5g top), they were part of a modular cluster (Supplementary Fig. 10a, black boxes on the left, fifth square from above). We also performed kernel-weighted density estimation (Supplementary Fig. 10b), confirming our results that *Fgf7* and *Pappa* were co-expressed in the

SASP cells. Likewise, *Dkk1* and *Cdkn2a/p16^Ink4a* displayed the mathematically predicted comparable expression patterns in kernel-weighed density, displayed in tSNE, as did *Bmp2* and *Cdkn1a/p21^Cip1* (Fig. 5g, Supplementary Fig. 10b).

## Further experimental validation of in silico analyses

The above analyses of both hematopoietic and mesenchymal scRNA-seq data pointed to *Mif* as a key SASP gene that should increase with senescent cell burden and be reduced following clearance of senescent cells. Thus, as a final validation of our in silico analyses, we examined *Mif* mRNA levels by RT-qPCR in our mouse models and found that as predicted, *Mif* mRNA levels were increased in the bones from old compared to young mice (Fig. 6a) and were significantly reduced

**Fig. 4 | SASP-associated hematopoietic cells in human bone marrow are mainly of monocytic origin and communicate via the MIF pathway. a** Using a previously published scRNA-seq dataset from human bone marrow (GSE120446[48], n = 68,478 cells), we performed GSEA at the single cell level to uncover cells responsible for senescence/SASP-associated gene expression. The highest enrichment score (ES) for the SenMayo gene set (purple) occurred within the CD14[+] and CD16[+] monocytic cell cluster, represented in a Uniform Manifold Approximation and Projection (UMAP). We selected the top 10% of senescence/SASP-expressing cells to form the "SASP cells" (n = 6850 cells) cluster displaying an (**b**) independent enrichment of canonical senescence genes including *CDKN1A/p21^Cip1* and *TGFB1* and which was likewise enriched for two aging signatures (GenAge: genes associated with aging in model organisms;[49] and CellAge: positively regulated genes associated with aging in human cells (SASP cells are marked purple). *T*-test with adjustment for multiple testing according to the hurdle model from MAST package (*CDKN1A*: p < 0.0001; *TGFB1*: p < 0.0001). **c** The SASP cells showed the highest interaction strength with T cells in the bone marrow, the numbers represent the relative interaction strength as sum of interaction weights. Edge weights are proportional to interaction strength, and a thicker line refers to a stronger signal [52]; **d** Among the interaction targets of SASP cells, T cells were predominantly targeted via the MHC-I, MIF, and PECAM1 pathways; **e** Members of the MIF and PECAM1 signaling pathways showed high expression patterns within the SASP population; **f** SASP cells were characterized by distinct co-expression patterns predicting functional clusters (e.g., *JUN* and *CDKN2A*), potentially overcoming difficulties of low expression of specific senescence-associated genes such as *CDKN2A/P16^ink4A*. These strong indicators of co-expression were mathematically isolated by z-scores (Spearman correlation) (**g**) and spatially summarized (**h**) in sub-cell populations within the SASP cluster, as indicated by kernel gene-weighted density estimation in a t-distributed Stochastic Neighbor Embedding (tSNE) representation (*EREG–IL1B*: p < 0.0001, *ICAM1–CDKN1A*: p < 0.0001, *JUN–CDKN2A*: p < 0.0001). ****p < 0.0001, n = 22 (10 ♂, 12 ♀). The error bands show a confidence interval level of 0.95. Boxplot minimum is the smallest value within 1.5 times interquartile range below 25th percentile, maximum is the largest value within 1.5 times interquartile range above 75th percentile. Centre is the 50th percentile (median), box bounds 25th and 75th percentile. Source data are provided as a Source Data file.

following the genetic clearance of senescent cells with AP in old *INKATTAC* mice (Fig. 6b).

## Discussion

The identification and characterization of senescent cells, particularly in bulk or scRNA-seq data, has been problematic for a number of reasons, including variable detection of low levels of the *Cdkn2a/p16^Ink4a* transcript even in senescent cells[47] and the lack of a consistent gene panel that can reliably identify these cells. Thus, we generated a gene set (SenMayo) consisting of 125 previously identified senescence/SASP-associated factors and first validated it in bone biopsy samples from two human cohorts consisting of young vs elderly women[27,28]. Importantly, to establish this as a senescence, rather than just "aging" gene set, we demonstrated that clearance of senescent cells in mice and in humans resulted in significant reductions of SenMayo. Using publicly available RNA-seq data, we demonstrated applicability across tissues and species and also found that SenMayo performed better than six existing senescence/SASP gene panels[14,42–46]. We next applied SenMayo to publicly available bone marrow/bone scRNA-seq data and successfully characterized hematopoietic and mesenchymal cells expressing high levels of senescence/SASP markers at the single cell level, demonstrated co-expression (where feasible) with the key senescence genes, *Cdkn2a/p16^Ink4a* and *Cdkn1a/p21^Cip1*, and analyzed intercellular communication patterns of senescent cells with other cells in their microenvironment. Based on these analyses, we found that senescent hematopoietic and mesenchymal cells communicated with other cells through common pathways, including the Macrophage Migration Inhibitory Factor (MIF) pathway, which has been implicated not only in inflammation but also in immune evasion, an important property of senescent cells[55]. Finally, as a key validation of our in silico analyses, we then examined *Mif* mRNA levels by RT-qPCR in our mouse models and found that as predicted, *Mif* mRNA levels were increased in bones from old compared to young mice and were significantly reduced following the genetic clearance of senescent cells in the old mice.

The heterogeneous composition of the SASP, which consists of a multitude of growth factors, chemokines, cytokines, and matrix-degrading proteins, has been experimentally verified using various in vitro systems to induce cell stress, in vivo using multiple pre-clinical animal models of aging and disease, and has been linked to several pathophysiological conditions in humans as well as clinical outcomes[56,57]. In the current study, we were able to group these factors into 9 distinct clusters to form tightly connected networks with distinct key molecules. The importance of these and other SASP factors has been verified in multiple biological contexts[58–65]. Interestingly, the control of the SASP itself by RELA/p65, which we detected in two sequencing datasets of aging women, has recently been experimentally verified in U2OS osteosarcoma cells[66].

Transcriptome-wide state-of-the-art technologies such as scRNA-seq will help shape our understanding of not just aging, but also therapeutics that potentially target fundamental mechanisms of aging, such as senolytics. As noted earlier, a confounder in these analyses is the generally low expression of the canonical marker of senescence, *Cdkn2a/p16^Ink4a*, which is clearly detectable by RT-qPCR in the setting of aging, but poses challenges when using transcriptome-wide approaches[47]. Hence, we propose a species-specific co-expression analysis with *JUN* (*Homo sapiens*) or *Dkk1* (*Mus musculus*), based on modules of comparable expression to address this challenge. To our knowledge, we for the first time leveraged publicly available single cell datasets to enrich for a senescence/SASP gene set. Since we did not include commonly used senescence-markers (*Cdkn2a/p16^Ink4a*, *Cdkn1a/p21^Cip1*) in the SenMayo panel, we were still able to rely on them to confirm a senescent cell state. Additional verification included a shift in the cell cycle phase to G1, as senescence prevents cells from proceeding to the S or M phases[41,61,67]. With *Cdkn1a/p21^Cip1* being expressed at relatively higher levels, we were able to verify a senescent status of SASP cells, confirming our approach to identify single cells expressing high levels of SenMayo genes as being senescent.

The use of pseudotime in scRNA-seq datasets to predict age-associated changes and fate commitment has been demonstrated previously in muscle stem cells (MuSCs) and fibro-adipose progenitors (FAPs)[68]. These analyses pointed to the importance of TGF-β signaling, but without specifically focusing on age-related expression changes. By contrast, we used pseudotime analyses to establish an innovative approach to identify age-dependent transcriptional changes in senescence/SASP genes distinct from *Cdkn2a/p16^Ink4a* and *Cdkn1a/p21^Cip1*.

Using a z-score based probabilistic model with pairwise correlations (bigSCale[69]) to construct transcriptional networks, several groups have successfully established the use of within-cell networks in single cell datasets[25,70] and we made use of this approach to define senescence modules of similar expression. With overall agreement between pseudotime, network analyses, and direct pairwise z-score prediction, we overcame the downside of normalized expression, and a z-score predicted space allowed us to assign clusters and spatially depict them within cellular aggregates. These modules may serve as sources for senescent markers and pathways[71].

As noted earlier, the MIF pathway emerged as a key intercellular communication pathway used by both hematopoietic and mesenchymal cells in bone marrow expressing high levels of senescence/SASP genes. This is perhaps not surprising given the importance of MIF as a pro-inflammatory cytokine, inhibitor of p53, and positive regulator of NF-κB[72]. MIF appears to be pivotal for cellular senescence, aging, and joint inflammation; however, its presence has been associated with a beneficial effect on the healthy lung and in MSCs[73–78]. Of note, recent evidence indicates an important role for MIF signaling in immune

**Table 2 | Top 20 significantly upregulated genes in the human and murine SASP clusters. Multiple *t*-test with Benjamini–Hochberg adjustment**

| Gene | avg_log2FC | Adj. *p* value |
|---|---|---|
| Human | | |
| S100A9 | 1.974317678 | 0 |
| CXCL8 | 1.817775253 | 0 |
| CST3 | 1.813835295 | 0 |
| TYROBP | 1.742773952 | 0 |
| LST1 | 1.704456515 | 0 |
| FCN1 | 1.704148119 | 0 |
| FCER1G | 1.698071186 | 0 |
| LYZ | 1.695879021 | 0 |
| CCL3 | 1.68109761 | 0 |
| S100A8 | 1.639167524 | 0 |
| CTSS | 1.605107533 | 0 |
| AIF1 | 1.537560282 | 0 |
| S100A12 | 1.501013381 | 0 |
| SAT1 | 1.475740324 | 0 |
| G0S2 | 1.471768259 | 0 |
| S100A11 | 1.426583167 | 0 |
| PSAP | 1.412156019 | 0 |
| NEAT1 | 1.402008889 | 0 |
| CSTA | 1.346171061 | 0 |
| SERPINA1 | 1.343012763 | 0 |
| Murine | | |
| Ccl2 | 1.385385456 | 4.4042E-274 |
| Cxcl14 | 1.348765531 | 0 |
| Cxcl12 | 1.348099221 | 0 |
| Hp | 1.32967138 | 2.6772E-298 |
| Trf | 1.32483506 | 6.8972E-280 |
| Serping1 | 1.304871038 | 0 |
| Mt1 | 1.294617837 | 0 |
| Tmem176b | 1.238330075 | 0 |
| Mt2 | 1.224158694 | 0 |
| Igfbp4 | 1.210905773 | 0 |
| Grem1 | 1.207056724 | 0 |
| Cd302 | 1.195220747 | 0 |
| Apoe | 1.163062993 | 0 |
| Msmp | 1.16240244 | 3.2104E-194 |
| Adipoq | 1.140888365 | 7.4114E-283 |
| Cyr61 | 1.136426625 | 0 |
| Gas6 | 1.110474329 | 0 |
| Mmp13 | 1.095488296 | 0 |
| Tmem176a | 1.087765581 | 0 |
| sCol3a1 | 1.082720154 | 1.1707E-254 |

evasion by tumors[79] and parasites[80], raising the possibility that increased MIF expression by multiple senescent cell types may play a role in the ability of senescent cells to resist immune clearance, particularly with aging[55], and this possibility warrants further study. Importantly, we also used *Mif* expression to validate our in silico predictions based on the scRNA-seq analyses, and confirmed both an increase in *Mif* expression with aging in murine bone as well as a reduction in *Mif* mRNA levels following genetic clearance of senescent cells.

The development and validation of SenMayo, as demonstrated here, may be particularly timely in the context of the recent establishment of a major NIH Common Fund consortium to map senescent

cells (SenNET, https://sennetconsortium.org/). The goal of this program is to "comprehensively identify and characterize the differences in senescent cells across the body, across various states of human health, and across the lifespan." The application of SenMayo to bulk or scRNA-seq data from SenNET should greatly facilitate this goal and could provide a standardized gene set that is used across the multiple sites involved in this consortium.

In summary, our studies contribute a novel gene set (SenMayo) that increases with aging across tissues and species, is responsive to senescent cell clearance, and can be used both in bulk and scRNA-seq analyses to identify cells expressing high levels of senescence/SASP genes. This gene set also has potential utility in the clinical evaluation of senescent cell burden and for studies of senolytic therapies. In addition, SenMayo circumvents current limitations in the transcriptional identification of senescent cells at the single cell level, thereby allowing for detailed analyses (e.g. pseudotime, intercellular signaling) that will facilitate better characterization of these cells in future studies.

## Methods
All research complies with the Declaration of Helsinki, and the study protocols were approved by the the Mayo Clinic Institutional Review Board. All subjects provided written, informed consent prior to enrolling in the studies. In order to protect participant privacy, we only provide composite data (e.g., mean ± SD for age) and individual data is entirely anonymized (i.e., does not include the sex or age of the individual participant).

### Generation of SenMayo
Our own GSEA gene set for senescence-associated genes was generated by combining genes that had been reported in previous studies to be enriched in senescent and/or SASP-secreting cells and experimentally verified in at least human or mouse cells. We screened 1,656 studies, but following removal of studies reporting duplicates, case reports, and non-human or non-murine genes, formulated a list of 15 studies from which we identified 125 genes that constituted SenMayo (Supplementary Data 1[18,26,50,57,63,81–90]). Note that we intentionally did not include *CDKN2A/p16^{Ink}* or *CDKN1A/p21^{Cip1}* in SenMayo as we used these genes, in part, to validate our senescence/SASP gene set. Likewise, and to not bias the subsequent analyses towards NF-κB-dependent SASP members, we excluded key regulatory factors like RELA and NF-κB1.

*RNA-seq.* Transcriptome-wide gene expression data from two independent cohorts of young and older postmenopausal women previously studied by our group (Cohort A, young [n = 19, 30.3 ± 5.4 years] and postmenopausal [n = 19, 73.1 ± 6.6 years]; and cohort B, young [n = 15, 30.9 ± 4.0 years] and postmenopausal [n = 15, 68.7 ± 4.8 years]) as well as 9 diabetic kidney disease patients (7 male, 2 female, 68.7 ± 3.1 years) were analyzed from three previous studies performed by our group (GSE141595: NCT02554695, GSE72815: NCT02349113:[24,39], NCT02848131)[27,28]. This was an open label Phase 1 pilot study[24], in which three days of oral dasatinib (100 mg/day) and quercetin (1000 mg/day) were administered to patients with diabetic kidney disease (ClinicalTrials.gov NCT02848131). Written, informed consent was obtained from all study participants. The primary outcome was a change in proportion of senescent cells present, and SenMayo is part of an exploratory analysis. Note that because this was a pilot, "proof-of-concept" study that was the first of its kind to use dasatinib+quercetin in humans, the investigators believed that an interim analysis to define potential efficacy of dasatinib+quercetin in humans was justified; indeed, plans for an interim analysis were included in the original protocol. From these subjects, we analyzed adipose tissue biopsies taken before and 11 days after completing the senolytic treatment. All human studies were approved by the Mayo Clinic Institutional Review Board and written informed consent was obtained from all participants. RNA was isolated from whole bone biopsies (which included

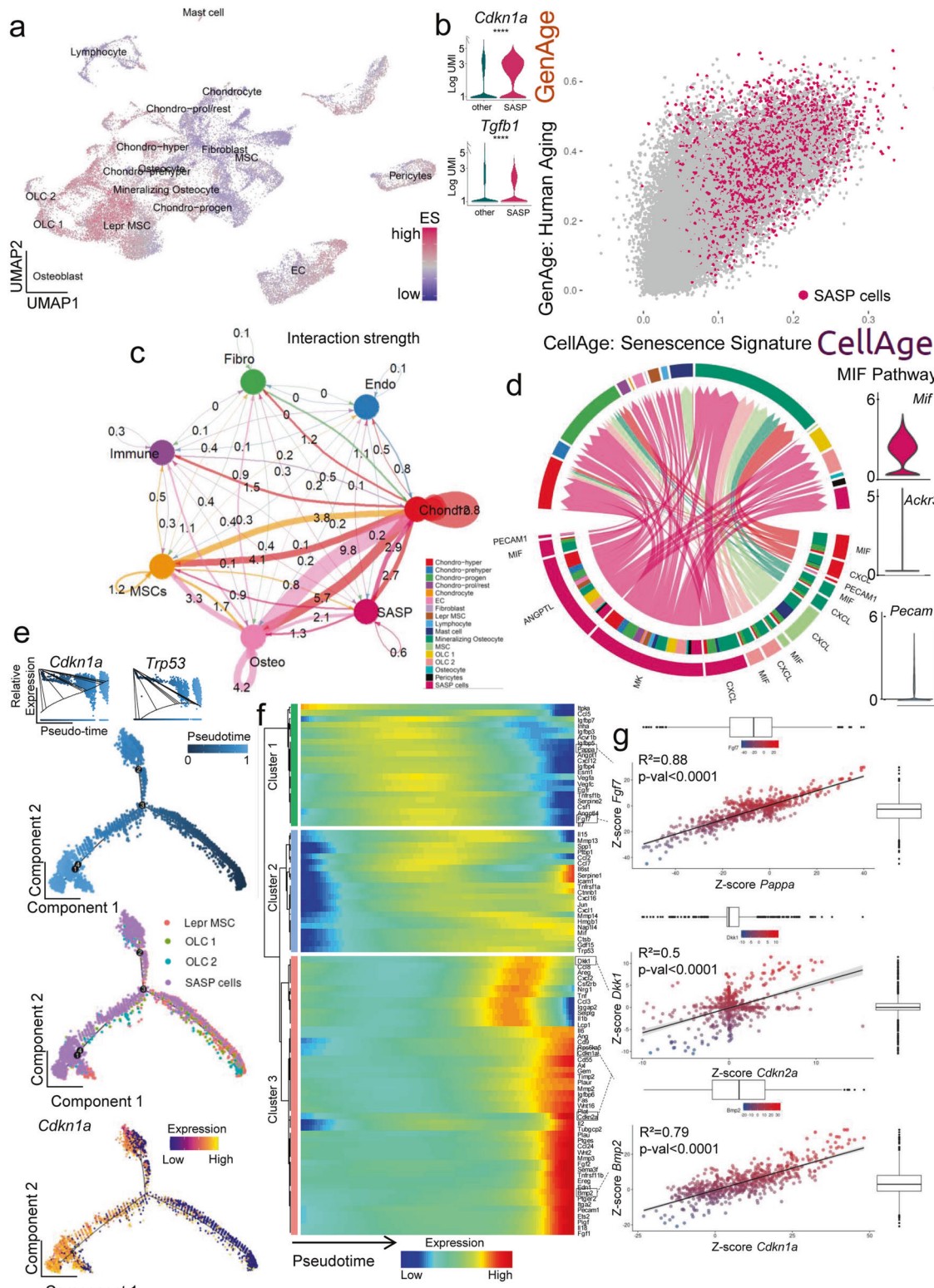

bone and bone marrow cells, Cohort A)[27] as well as bone biopsies that were processed to remove bone marrow and bone surface cells and were thus highly enriched for osteocytes (Cohort B)[28], and adipose tissue, 2–5 cm inferior to the navel (for details, see[39]). Subcutaneous adipose tissue was obtained by an elliptical incisional biopsy at a point to the right or left, and 2–5 cm inferior to the navel[24,39]. Sequencing was performed on a HiSeq2000 (Illumina®), fastq files were mapped to the human reference genome hg19, and analysis was performed as previously described[27,28]. Significantly differentially regulated genes were

selected by a Benjamini–Hochberg adjusted $p$ value <0.05 and log$_2$-fold changes above 0.5 or below −0.5. Gene Set Enrichment Analysis (GSEA[32,91]) was performed with default settings (1000 permutations for gene sets, Signal2Noise metric for ranking genes). The network analysis was conducted with Cytoscape 3.8.2 and the plugin iRegulon[30,92]. For mRNA-seq of murine material, tibiae were centrifuged as noted above to remove bone marrow elements and then were immediately homogenized in QIAzol Lysis Reagent (QIAGEN, Valencia, CA) and stored at −80 °C, until the time of RNA extraction. RNA-sequencing

**Fig. 5 | In murine bone and bone marrow mesenchymal cells, osteolineage cells constitute the largest proportion of SASP cells and communicate with osteolineage and chondrocytic cells via the MIF and PECAM1 pathways and show characteristics of terminal differentiation. a** We analyzed a publicly available murine bone and bone marrow gene set (GSE128423[54]), and enriched 35,368 cells for the SenMayo gene set; **b** The top 10% senescence/SASP gene-expressing cells ($n = 3537$) were assigned to the "SASP cells" cluster. They displayed an increase in canonical markers of senescence including *Cdkn1a/p21*[Cip1] and *Tgfb1*, and were enriched in the GenAge and CellAge gene sets (GenAge, CellAge[49]); *T*-test with adjustment for multiple testing according to the hurdle model from MAST package (*Cdkn1a*: $p < 0.0001$, Tgfb1: $p < 0.0001$). **c** The strongest interaction of the SASP cells was narrowed down to chondrocytic cells, while the osteolineage cells were another important crosstalk neighbor, the numbers represent the relative interaction strength as sum of interaction weights. Edge weights are proportional to interaction strength, and a thicker line refers to a stronger signal[52]. Two-sided unpaired *t*-test except for CCL2 in Cohort A: Kolmogorov–Smirnov test. **d** Outgoing interaction patterns of SASP cells (pink, left bottom quarter) indicated the

importance of several signaling pathways that resulted in a significant enrichment of Mk, Angptl, Mif and Pecam1; (**e**) In pseudotime, the SASP cluster was most abundant in the terminal branches, and overexpressed Cdkn1a/p21[Cip1] in terminal states (top-left inlay: the solid line represents the expression values as a function of pseudotime-progress, bottom red color on the left, terminal branch); **f** In their terminal differentiation, the SASP cluster was enriched in several factors, out of which distinct co-expressional patterns were extracted (Spearman correlation); **g** While the terminal differentiation was marked by a simultaneous loss of Pappa and Fgf7 (cluster 1, green in **f**), a significant correlation of Dkk1 with Cdkn2a/p16[Ink4l], likewise Bmp2 and Cdkn1a/p21[Cip1], was mathematically predicted (cluster 2, pink in f). Fgf7–Pappa: $p < 0.0001$, Dkk1–Cdkn2a: $p < 0.0001$, Bmp2–Cdkn1a: $p < 0.0001$. ****$p < 0.0001$, $n = 8$ (4 bone, 4 bone marrow, all male). Depicted are mean ± SEM. The error bands show a confidence interval level of 0.95. Boxplot minimum is the smallest value within 1.5 times interquartile range below 25th percentile, maximum is the largest value within 1.5 times interquartile range above 75th percentile. Centre is the 50th percentile (median), box bounds 25th and 75th percentile. Source data are provided as a Source Data file.

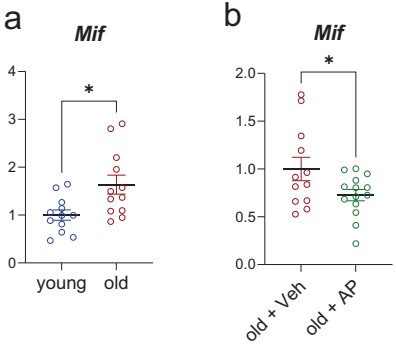

**Fig. 6 | The in silico predicted importance of the Mif pathway is reflected in the aged INK-ATTAC mouse model. a** We compared young ($n = 12$) and old vehicle-treated mice ($n = 13$), and old mice treated with AP ($n = 16$). **a** Upregulation of *Mif* was confirmed by RT-qPCR ($n = 24$ young [12 Veh, 12 old (all female), $p = 0.0102$]); **b** The clearance of senescent cells in the aged cohort by AP treatment reduced this *Mif* expression ($n = 26$ old [12 Veh, 14 AP, all female], $p = 0.0459$). *$p < 0.05$. Two-sided unpaired *t*-tests. Source data are provided as a Source Data file.

was performed on a HiSeq2000 (Illumina®), fastq files were mapped to the murine reference genome mm10, and analysis was performed as previously described[27,28]. An example of the code used for RNA-seq can be found in the provided R notebook (Methods: GSE72815_YOE_Notebook.txt).

### Mouse strains and drug treatments

All animal protocols were approved by the Institutional Animal Care and Use Committee (IACUC), and all experiments were performed in accordance with IACUC guidelines. Mice were housed in ventilated cages in a pathogen-free facility (12-hour light/dark cycle, 23 °C) and had access to food (standard mouse diet, Lab Diet 5053, St. Louis, MO) and water *ad libitum*. Mouse experiments for a genetic targeting approach of senescent cells have been described by our group earlier[37]. Briefly, old (20 months) female mice were injected intraperitoneally with vehicle (4% of 100% EtOH, 10%PEG400, 86% of 2% Tween 20 in deionized Water) or AP20187 (B/B homodimerizer, Clontech; 10 mg of AP20187 *per* kg body mass) twice weekly at the age of 20 months for a total of 4 months (old mice were sacrificed at 24 months of age). In addition, young (6-month) *INK-ATTAC* mice were used as a control comparison cohort.

### Quantitative real-time polymerase chain reaction (RT-qPCR) analysis

For bone analyses, tibiae were centrifuged to remove marrow elements and then immediately homogenized in QIAzol Lysis Reagent

(QIAGEN, Valencia, CA) and stored at −80 °C. Subsequent RNA extraction, cDNA synthesis, and targeted gene expression measurements of mRNA levels by RT-qPCR were performed[93] Total RNA was extracted according to the manufacturer's instructions using QIAzol Lysis Reagent. Purification with RNeasy Mini Columns (QIAGEN, Valencia, CA) was subsequently performed. On-column RNase-free DNase solution (QIAGEN, Valencia, CA) was applied to degrade contaminating genomic DNA. RNA quantity was assessed with Nanodrop spectrophotometry (Thermo Fisher Scientific, Wilmington, DE). Standard reverse transcriptase was performed using High-Capacity cDNA Reverse Transcription Kit (Applied Biosystems by Life Technologies, Foster City, CA). Transcript mRNA levels were determined by RT-qPCR on the ABI Prism 7900HT Real Time System (Applied Biosystems, Carlsbad, CA) using SYBR green (Qiagen, Valencia, CA). The mouse forward primer sequence (5′–3′) for *Mif* was: 5′-GCCACCATGCCT ATGTTCATC-3′ and Reverse Primer Sequence 5′-GGGTGAGCTC CGACAGAAAC-3′. RNA was normalized using two reference genes (*Actb* [forward: 5′-AATCGTGCGTGACATCAAAGAG-3′, reverse: 5′-GCCATCTCCTGCTCGAAGTC-3′], *Gapdh* [forward: 5′-GACCTGACCTGC CGTCTAGAAA-3′, reverse: 5′-CCTGCTTCACCACCTTCTTGA-3′]) from which the most stable housekeeping gene was determined by the geNorm algorithm. For each sample, the median cycle threshold (Ct) of each gene (run in triplicate) was normalized to the geometric mean of the median Ct of the most stable reference gene. The delta Ct for each gene was used to calculate the relative mRNA expression changes for each sample. Genes with Ct values >35 were considered not expressed (NE), as done previously[94].

### Single-cell RNA-seq (scRNA-seq) analysis

Transcriptome-wide analysis of human bone marrow mononuclear cells at a single cell level was based on a previously published dataset[48]. Here, bone marrow was isolated from healthy female ($n = 10$) and male ($n = 10$) donors ($50.6 \pm 14.9$ years) and droplet-based scRNA-seq was performed. A minimum sequencing depth of 50,000 reads/cell with a mean of 880 genes/cell was reported. In addition, we analyzed droplet-based scRNA-seq data from bone marrow cells isolated from C57BL/6 mice ($n = 14$)[54] and from C57BL/6JN mice ($n = 30$)[36] and from the *tabula muris senis*[36]. Sequencing data were aligned to the human reference genome Grch38 and the mouse genome mm10, respectively. Data with at least 500 unique molecular identifiers (UMIs), log10 genes per UMI >0.8, >250 genes *per* cell and a mitochondrial ratio of less than 20% were extracted, normalized, and integrated using the Seurat package v3.0 in R4.0.2. After quality control and integration, we performed a modularity optimized Louvain clustering with the resolution "1.4", leading to 40 distinct clusters in the human dataset. Subsequently, we performed the labelling for these 40 clusters manually with established key marker genes (Supplementary Fig. 3a). In the murine

dataset, we chose the same order of analysis, and picked the resolution "1.4", leading to 40 distinct clusters, which were manually assigned according to established marker genes (Supplementary Fig. 6a).

Subsequent R-packages were Nebulosa (3.13[95]), Monocle (2.18.0[96]), dittoSeq (1.2.6[97]), Escape (1.0.1, "Borcherding N, Andrews J (2021). escape: Easy single cell analysis platform for enrichment. R package version 1.2.0."), Cellchat[52] (within the Cellchat package, and for Fig. 4c we aimed to summarize functional cell types in order to highlight the functional importance of SASP cells and their communicational patterns. Subsequently, we combined "CD10+ B cells", "CD20+ B cells", "Plasma cells", "Plasmacytoid dendric cells", "Conventional dendric cells" as "B cells", "CD4+ naïve T cells", "CD4+ memory T cells", "CD8+ naïve T cells", "CD8+ effector T cells" were summarized as "T cells", "Early erythroid progenitors", "Early erythrocytes", "Late erythrocytes" as "Ery", "HSPCs" as "HSPCs", "Monocyte progenitors", "CD14+ monocytes", "CD16+ monocytes", "Macrophages", "Natural killer cells" as "Mono" and "SASP cells" as "SASP". For Fig. 5c, "Chondro-hyper", "Chondro-prehyper", "Chondro-progen", "Chondro-prol/rest", "Chondrocyte" were summarized as "Chondro", "EC", "Pericytes" as "Endo", "Fibroblast" as "Fibro", "Lymphocyte", "Mast cell" as "Immune", "Lepr MSC", "MSC" as "MSC", "Mineralizing Osteocyte", "OLC 1", "OLC 2", "Osteoblast", "Osteocyte" as "Osteo" and "SASP cells" as "SASP"), bigSCale (2.1[71]), gprofiler2 (0.2.0[98]), igraph (1.2.6, Csardi G, Nepusz T (2006). "The igraph software package for complex network research." InterJournal, Complex Systems, 1695), PCAtools (2.4.0, Blighe K, Lun A (2021). PCAtools: PCAtools: Everything Principal Components Analysis. R package version 2.4.0), and corrplot (0.89) and SCENIC (1.2.4[99]).

The gene ontology and KEGG analyses (Supplementary Figs. 3f, S7d) with gprofiler2 were done for homo sapiens (hsapiens) and mus musculus (mmusculus) after the positively regulated genes were selected with the "FindMarkers" function (used test = "MAST", logfc.threshold = 0.25). For the gost function, we used an user threshold of 0.05 and fdr correction method.

Pseudotime is a progression of cells along a virtually estimated path, mimicking temporal development. By using Monocle, an independent component analysis (ICA) dimensional reduction is performed, followed by a projection of a minimal spanning tree (MST) of the cell's location in this reduced space. Each cell is assigned a pseudotemporal space[100,101]. Monocle 2 was used to preprocess, perform UMAP reduction, and reduce the dimensionality using the DDRTree algorithm with a maximum of four dimensions. Subsequently, the cells were ordered and genes plotted along the reduced dimension. Differential gene testing has been performed with the formula "~sm.ns(Pseudotime)", and the results were restricted by a $q$ value <0.1[100].

An example of the code used for scRNA-seq can be found in the provided R notebook (Methods: R_notebook_Fig4_5_sup2to5.Rmd).

### Reporting summary

Further information on research design is available in the Nature Research Reporting Summary linked to this article.

## Data availability

The data supporting the findings from this study are available within the manuscript and its supplementary information. The RNA-seq data from Fig. 1 have been deposited in the National Center for Biotechnology Information's Gene Expression Omnibus under GSE72815) and GSE141595. The RNA-sequencing data from Mus musculus brain microglia (Fig. 2a) is deposited under (GSE145265), prefrontal cortex (Fig. 2b) under GSE128770, dorsal hippocampus (Fig. 2c) under GSE94832 and the tabula muris senis (Fig. 2d) under GSE149590). The human single cell sequencing data (Fig. 4) is stored at GSE120221, while the murine single cell sequencing data (Fig. 5) is stored at GSE128423. The murine INK-ATTAC tibia diaphysis bulk RNA-

sequencing data (Fig. 3a and b) is available from dryad (https://datadryad.org/stash/share/YdD6C2ZFDgSizXehPR0qqPy4io7oRQJM-GRlrPuij9WU) or GSE199493. The human subcutaneous fat bulk RNA-sequencing data from trial no. NCT02848131 (Fig. 3d and e) is available from dryad (https://datadryad.org/stash/share/YdD6C2ZFDgSizX-ehPR0qqPy4io7oRQJMGRlrPuij9WU) and PRJNA826433. Source data are provided with this paper.

## Code availability

To reproduce the analyses, several notebooks are included. For the RNA-sequencing datasets, an RNA-seq R notebook leads the reader through the RNA-seq analyses: "Supplementary Code 1" (https://datadryad.org/stash/share/YdD6C2ZFDgSizXehPR0qqPy4io7oRQJMGRlrPuij9WU). In order to acquire the single cell datasets, the notebook "Supplementary Code 2" has been designed (https://datadryad.org/stash/share/YdD6C2ZFDgSizXehPR0qqPy4io7oRQJMGRlrPuij9WU). For the single cell analyses (Figs. 4 and 5, supplementary Figs. 2–10), the notebook "Supplementary Code 3" provides the figures' underlying code (https://datadryad.org/stash/share/YdD6C2ZFDgSizXehPR0qqPy4io7oRQJMGRlrPuij9WU). All of these notebooks can be downloaded from dryad (https://datadryad.org/stash/share/YdD6C2ZFDgSizXehPR0qqPy4io7oRQJMGRlrPuij9WU).

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

## Acknowledgements

This work was supported by the German Research Foundation (D.F.G., 413501650) (D.S.), National Institutes of Health (NIH) grants P01 AG062413 (S.K., J.N.F., N.K.L., R.P., P.D.R., L.J.N., Y.I., J.P., D.G.M., T.T., J.L.K.), R01 AG076515 (S.K., D.G.M.), R21 AG065868 (S.K., J.N.F), K01 AR070241 (J.N.F.), R01 AG063707 (D.G.M.), R37 AG 013925 (J.L.K., T.T.), R33AG 61456 (J.L.K., T.T., R.P., P.D.R., L.J.N., S.K.), 1R01AG068048-01 (JFP), R56 AG 60907 and R01 AG55529 (N.K.L.)., the Connor Fund (J.L.K., T.T.), Robert P. and Arlene R. Kogod (J.L.K.), Robert J. and Theresa W. Ryan (J.L.K., T.T.), the Noaber Foundation (J.L.K., T.T.), and Mildred Scheel postdoc fellowship by the German Cancer Aid (R.L.K.). X.Z. is supported by the Robert and Arlene Kogod Center on Aging Career Development Award. The authors thank SA Johnsen and FH Hamdan for inspiring discussions.

## Author contributions

D.S., J.N.F., and S.K. conceived and directed the project. D.S. and J.N.F. designed the experiments and interpreted the data with input from S.K. Experiments were performed by D.S., R.L.K., and M.L.D. D.S. and S.K. wrote the manuscript. E.J.A. and X.Z. contributed to the statistical/bioinformatic analyses and reviewed the manuscript. L.J.H., T.T., and J.L.K. oversaw the clinical trial involving D + Q and contributed data from the trial. A.X. was responsible for collecting and processing the samples from the clinical trial. N.K.L., R.J.P., P.D.R., L.J.N., Y,I., D.J., J.F.P., and D.G.M. contributed to the conceptual development of the project and input into the SenMayo geneset. All authors reviewed the manuscript. J.N.F. and S.K. oversaw all experimental design, data analyses, and manuscript preparation. J.N.F., S.K., and D.S. accept responsibility for the integrity of the data analysis.

## Competing interests

J.L.K., T.T., and N.K.L. have a financial interest related to this research. Patents on senolytic drugs and their uses and SASP biomarkers are held by Mayo Clinic and the University of Minnesota. This research has been reviewed by the Mayo Clinic Conflict of Interest Review Board and was conducted in compliance with Mayo Clinic Conflict of Interest policies. The remaining authors declare no competing interests
