## [Peer Review File · Nature Communications]

A New Gene Set Identifies Senescent Cells and Predicts Senescence-Associated Pathways Across TissuesReviewers' Comments:

Reviewer #1:

Remarks to the Author:

There are currently several suggested gene sets the expression of which is proposed to correlate with senescence of the cell population under study. This paper proposes another such set designated "SenMayo" and compares this with 6 other published gene sets, suggesting its superiority and applying it several human and mouse tissues of different sources before and after treatment with senolytics. SenMayo was derived for screening 1,656 studies resulting in the identification of 125 genes from 15 studies after excluding duplicates, case reports, and non-human or non-murine genes. SenMayo intentionally excluded p16 and p21 so that these could be used independently to validate detection of senescent cells. The SenMayo gene set was then tested in several different scenarios, concluding that its expression was higher in samples from older humans and mice across different tissues and decreased after senolytic treatment, and that it could be used on data from bulk and scRNA-seq analyses to identify cells expressing high levels of senescence/SASP genes. The strength of the work lies in the several different conditions investigated but it could be made a little clearer and easier on the reader to decipher what the cohorts are. The M&M lists "young (n=15, 30.9±4.0 years and n=19, 30.3±5.4 years) and postmenopausal females (n=15, 68.7±4.8 years and n=19, 73.1±6.6 years)". Please state here the difference between the two groups of young and older, and clarify their relationships with cohort A and B without requiring the reader to search intensively. I think it is also a far stretch to refer to the differences in gene expression between younger and older subjects in these cohorts as being informative or not for the "aging process". These differences reflect the presence of SASP+ cells in different proportions in cross-sectional comparisons and in tissues as diverse as brain, bone and hematopoietic cells, but do not explicitly reflect an aging process. Thus, I think it is something of a tautology to say SenMayo is associated "not only with aging but also specifically with cellular senescence" – in fact, only the latter is shown, and it is taken that that reflects the "aging process".

The paper begins with a single gene analysis of CDKN1A/p21Cip1, CCL2, IL6, NFKB1, RELA, and STAT3, yielding expected results in that samples from older women displayed an upregulation of these genes (but not CDKN2A/p16Ink4a). However, of these, only IL 6 is included in SenMayo. Please discuss why the others are not and whether IL 6 alone could really not substitute for all? The authors made the point that p21 and p16 were purposefully excluded. Was that the case with all these others as well, and if so, why? Other "canonical" markers are all included in SenMayo, eg. CCL24, SEMA3F, FGF2, and IGFBP7.

The authors argue for the superiority of SenMayo over other gene clusters and provide an example using Cohort A and reporting 2 of the 50 genes in R-HSA-2559582 that were significantly enriched in biopsies from older women, whereas 13 of the 120 SenMayo genes were, i.e. 4%-vs-10.8%, not such a big difference. What were the two R-HAS genes, were they contained in SenMayo, and is the expression pattern the same in different tissues and species?

In the M&M, there is no description of the phase I pilot study in which the senolytic combination Dasatinib plus Quercetin was used to treat patients with diabetic kidney disease. Although this is published elsewhere, to assist the reader, please describe in the M&M here.

Nonetheless, this is an impressive paper, especially the monitoring of gene expression in adipose tissue samples from people treated with Dasatinib plus Quercetin, as well as the broad range of consistent data in mouse and human, and the cross-talk experiments delineating inföammaging pathways. These data could have meaningful clinical consequences.

Reviewer #2:

Remarks to the Author:

In the manuscript entitled "A New Gene Set Identifies Senescent Cells and Predicts Senescence-Associated Pathways Across Tissues" the authors developed a new senescence gene set ("SenMayo") and validated GSEA analysis of SenMayo from humans, mice, and changes in this gene set following the clearance of senescent cells. Furthermore, the utility of SenMayo was demonstrated at the single cell level.

The technical quality of the work is high and supports the conclusions of the manuscript. The authors provide compelling and carefully controlled data from distinct model systems: human and mice; aging and senescent cell clearance. For improving this paper more, several points and suggestions are listed below.

Specific comments:

1. In methods - Single-cell RNA-seq (scRNA-seq) analysis and page 29, the authors define that "Plasmacytoid dendric cells", "Conventional dendric cells" were summarized as "B cells" combined with "CD10+ B cells", "CD20+ B cells", "Plasma cells". Why so? Authors should provide explanations.
2. For Fig. 4E & Fig.5D, there is a lack of fair comparison and evaluation of the gene expression about members of MIF and PECAM1 signaling pathway among SASP cells and other cell types.
3. The authors performed pseudotime analysis to demonstrate that the SASP cluster, in both bone marrow hematopoietic cells and bone/bone marrow mesenchymal cells, was in the final phases of cellular differentiation. Although, in Fig. 5E, authors coincidence the result with prior knowledge (the increased expression of Cdkn1a/p21Cip1 and Trp53), monocle cannot be used to determine the base state of the trajectory. Authors should use an unbiased trajectory method to confirm this.
4. The authors validated the changes of Mif mRNA levels in the bone of mouse model. However, the authors also provided an interesting conclusion showing that the MIF pathway has been important in both hematopoietic and mesenchymal cells to participate in cellular interactions. Authors should provide validation of this interesting interaction analysis.

Minor comments:

1. In methods - Single-cell RNA-seq (scRNA-seq) analysis and page 29, there exists a minor error. "For Fig. 2C," and "For Fig. 3C" should be "for Fig. 4C," and "For Fig. 5C", respectively.
2. For Fig. S2 B-C & Fig. S3E, the color legend needs to show the cell-type names, to avoid confusion about the interpretation of the data.

Reviewer #3:

Remarks to the Author:

This manuscript by Dominik Saul et al. generated a novel gene set (Sen Mayo) including previously reported genes that were enriched in senescent and/or SASP-secreting cells. For RNA-seq, transcriptome-wide gene expression data was obtained from 2 different cohorts and further employed these data to identify the senescent cell in vivo. Meanwhile, the author also validate their hypothesis in single-cell resolution, and combined with trajectory inference methods to explore senescence-associated pathways. Overall, this study provided comprehensive correlative data in a clearly written way, however, there are several concerns addressed as follows:

Major comments:

1. To determine whether senescence-and SASP-associated pathways were enriched with aging, the authors selected to analyze transcriptional regulatory relationships (Figure 1B). It would be very

interesting, if the author can discuss what happens if these transcription factors and regulate signaling molecules are perturbed? This might straightforwardly leverage some public databases and tools, including PMID: 32051003, 28991892, and 25058159. Meanwhile, please provide further explanation regarding how SenMayo encodes a dense network of different protein classes in a strong interaction network (Figure 1F). Tables of cluster analysis results can be additionally included as part of the supplementary section.

2. Since the senolytic drug combination, Dasatinib plus Quercetin (D+Q) in the context of diabetic kidney disease was used to validate the ability of SenMayo on predicting senescent cell clearance (Figure 3), please describe the concept for choosing diabetic kidney disease model to validate the data.

3. In the previous analysis, principal component analysis (PCA) was performed to estimate/evaluate the variance displayed in Figure 3C. What about UMAP variance estimation/evaluation? Further explanations are required to clarify the theories (Figure 4A).

4. How the cell types from single-cell transcriptomic data presented in Figures 4A and 5A were identified? Please explain in detail the methods for cell-type identification from single-cell transcriptomic data. Similar interpretation of scRNA-seq cell type annotation protocol presented in figures 4C and 5C. Since cell type annotation obtained from a previous study (PMID: 30518681) which is lack of canonical genes references for each cluster, should the new cluster referred as "SASP cells" generated by selecting top 10% highest expression of senescence/SASP-associated genes technically be aligned together in the whole UMAP representation? Besides that, please describe in more detail regarding the interaction values between SASP and the other particular cell types. The two aging signatures CellAge and GenAge and its significant correlation to SASP also need to be interpreted more clearly.

5. Please present the reconstruction and basis of pseudo-time analysis in a more detailed manner, along with the procedure/protocol of trajectory analysis on the monocle package. The three main origins for the SASP cluster, namely Lepr+ MSCs, OLC 1, and OLC 2, were depicted in pseudo time by applying trajectory inference. Why only canonical markers of senescence were reported in Figure 5B other than the result of TGFB1 in a terminal developmental branch?

6. To further test SenMayo in single-cell datasets and potentially contrast bone marrow hematopoietic cells to bone/bone marrow mesenchymal cells, a published murine dataset containing scRNA-seq data from bone and bone marrow mesenchymal cells was leveraged. Why the authors did not select the human bone marrow mesenchymal cells related dataset to avoid biological heterogeneity? Regarding the selection of canonical SASP markers or SASP/senescence markers, verification of existing references may also improve the reproducibility of the current study.

7. The author used Seurat algorithm to analyze 500 unique molecular identifiers (UMIs) and data were obtained based on the following thresholds: log10 genes per UMI >0.8, >250 genes per cell, a mitochondrial ratio of less than 20%. Since results from Sequencing Data Alignment often undergo considerable fluctuations due to modification in specific parameters and tuning. Therefore, without detailed description of how the algorithms were applied, it may seem very difficult to reproduce these results. For example, how did the algorithms result in the number of subgroups and objects in each analysis and how was it concluded that these subgroups are statistically robust? To avoid being cluttered with unnecessary details, the authors may provide a summarized list in either the Supplementary or Code Availability section.

Minor comments:

1. All images are presented in a high-resolution and professional manner, however, to be more intelligible without reference, figure legends should provide sufficient details (such as the meaning of including solid line, dashed line, dash-dotted line, dotted line...) corresponding to each component part.

2. In Figure S4D, it is unclear how GO/KEGG pathway enrichment is analyses. Please also make sure the color codes clarify in these figures and tables. Meanwhile, this would make sense to state as an advantage if the study had indeed identified a novel pathway or any interaction not previously known, either via bulk or single-cell seq data.

Reviewer 1

There are currently several suggested gene sets the expression of which is proposed to correlate with senescence of the cell population under study. This paper proposes another such set designated "SenMayo" and compares this with 6 other published gene sets, suggesting its superiority and applying it several human and mouse tissues of different sources before and after treatment with senolytics. SenMayo was derived for screening 1,656 studies resulting in the identification of 125 genes from 15 studies after excluding duplicates, case reports, and non-human or non-murine genes. SenMayo intentionally excluded p16 and p21 so that these could be used independently to validate detection of senescent cells. The SenMayo gene set was then tested in several different scenarios, concluding that its expression was higher in samples from older humans and mice across different tissues and decreased after senolytic treatment, and that it could be used on data from bulk and scRNA-seq analyses to identify cells expressing high levels of senescence/SASP genes. The strength of the work lies in the several different conditions investigated but it could be made a little clearer and easier on the reader to decipher what the cohorts are.

1. *The M&M lists "young (n=15, 30.9±4.0 years and n=19, 30.3±5.4 years) and postmenopausal females (n=15, 68.7±4.8 years and n=19, 73.1±6.6 years)". Please state here the difference between the two groups of young and older, and clarify their relationships with cohort A and B without requiring the reader to search intensively.*

Response: We have clarified this point in the Methods (page 18). As now stated, both were independent cohorts of healthy young and older postmenopausal women we have previously studied. We have also clarified which groups corresponded to cohort A versus cohort B.

2. *I think it is also a far stretch to refer to the differences in gene expression between younger and older subjects in these cohorts as being informative or not for the "aging process". These differences reflect the presence of SASP+ cells in different proportions in cross-sectional comparisons and in tissues as diverse as brain, bone and hematopoietic cells, but do not explicitly reflect an aging process. Thus, I think it is something of a tautology to say SenMayo is associated "not only with aging but also specifically with cellular senescence" – in fact, only the latter is shown, and it is taken that that reflects the "aging process".*

Response: We acknowledge this point and have gone through the manuscript to dissociate, as much as possible, senescence from aging *per se*, as the Reviewer suggests.

3. *The paper begins with a single gene analysis of CDKN1A/p21Cip1, CCL2, IL6, NFKB1, RELA, and STAT3, yielding expected results in that samples from older women displayed an upregulation of these genes (but not CDKN2A/p16Ink4a). However, of these, only IL 6 is included in SenMayo. Please discuss why the others are not and whether IL6 alone could really not substitute for all? The authors made the point that p21 and p16 were purposefully excluded. Was that the case with all these others as well, and if so, why? Other "canonical" markers are all included in SenMayo, eg. CCL24, SEMA3F, FGF2, and IGFBP7.*

Response: We thank the reviewer for this comment. From the initial analysis, CDKN2A and CDKN1A were purposely excluded, while CCL2 and IL6 are members of the SenMayo. The transcription factor NF-κB family with RELA and NFKB1 are known regulators of the SASP, as is STAT3¹. Although some SASP members are indeed transcription factors, we aimed to exclude the higher-level regulators from the panel in order to not bias the subsequent analyses into one particular direction. Indeed, the findings in Fig. 1B drove us to further elucidate which SASP members would be affected. To clarify these insights for the reader, we have added "Likewise, and to not bias the subsequent analyses towards NF-κB-dependent SASP members, we excluded key regulatory factors like RELA and NF-κB1" to the Methods (page 18). Also, a summary of the

transcription factors that are important for SenMayo, and how they affect the RNA-seq as single cell RNA-seq data (making use of iRegulon and SCENIC), is now included in the new Supplementary Figures 1 and 8.

4. The authors argue for the superiority of SenMayo over other gene clusters and provide an example using Cohort A and reporting 2 of the 50 genes in R-HSA-2559582 that were significantly enriched in biopsies from older women, whereas 13 of the 120 SenMayo genes were, i.e. 4%-vs-10.8%, not such a big difference. What were the two R-HSA genes, were they contained in SenMayo, and is the expression pattern the same in different tissues and species?

Response: We thank the reviewer for the opportunity to clarify this. The two genes in R-HSA-2559582 were CDKN1A ($p_{adj}<0.001$) and IGFBP7 ($p_{adj}=0.036$). While the former was intentionally excluded from SenMayo, the latter is part of it. We agree that a 2/50 vs. 13/120 “hit rate” is not impressive, but want to point out the real strength of SenMayo, which results in a NES of 1.51 ($p=0.0023$) compared to R-HSA-2559582 with a NES of 1.11 ($p=0.2826$) for this cohort (Fig. 1 D and G). Also, as pointed out on page 6, the GSEA analysis includes not only genes that differ significantly between groups, but also evaluates overall trends for differences in gene expression between groups and hence provides considerably greater power than examining individual genes.

Regarding the expression pattern of CDKN1A and IGFBP7 across tissues and species we analyzed both of our single cell datasets for CDKN1A/IGFBP7 and Cdkn1a/Igfbp7 expression (Reviewer only. Fig. 1), demonstrating the high expression of both in the SASP population.

Reviewer only. Figure 1. Expression of CDKN1A/IGFBP7 in our human and Cdkn1a/Igfbp7 in our murine dataset. (A) CDKN1A shows a high expression in the SASP cells of our human bone marrow dataset, as does (B) IGFBP7. In the murine dataset, (C) Cdkn1a is highly expressed in

the SASP cells and EC, Fibroblast, Lepr⁺ MSCs and Pericytes, while (D) *Igfbp7* shows a similar pattern.

5. *In the M&M, there is no description of the phase I pilot study in which the senolytic combination Dasatinib plus Quercetin was used to treat patients with diabetic kidney disease. Although this is published elsewhere, to assist the reader, please describe in the M&M here.*

Response: We now provide additional details regarding this study, in addition to referring the reader to the original reference.

Nonetheless, this is an impressive paper, especially the monitoring of gene expression in adipose tissue samples from people treated with Dasatinib plus Quercetin, as well as the broad range of consistent data in mouse and human, and the cross-talk experiments delineating inflammaging pathways. These data could have meaningful clinical consequences.

Reviewer 2

In the manuscript entitled "A New Gene Set Identifies Senescent Cells and Predicts Senescence-Associated Pathways Across Tissues" the authors developed a new senescence gene set ("SenMayo") and validated GSEA analysis of SenMayo from humans, mice, and changes in this gene set following the clearance of senescent cells. Furthermore, the utility of SenMayo was demonstrated at the single cell level. The technical quality of the work is high and supports the conclusions of the manuscript. The authors provide compelling and carefully controlled data from distinct model systems: human and mice; aging and senescent cell clearance. For improving this paper more, several points and suggestions are listed below.

1. *In methods - Single-cell RNA-seq (scRNA-seq) analysis and page 29, the authors define that "Plasmacytoid dendric cells", "Conventional dendric cells" were summarized as "B cells" combined with "CD10+ B cells", "CD20+ B cells", "Plasma cells". Why so? Authors should provide explanations.*

Response: We thank the reviewer for the opportunity to clarify this. When performing a thorough analysis with all cell types and their communication patterns, we concluded that the overall message of CellChat was not conveyed as clearly as it would be if we combined the cell clusters into functional subunits (please see Reviewer only. Fig. 2), as the authors of CellChat suggest (² and <https://github.com/sqjin/CellChat>). This way, the message conveyed is more obvious and the importance of SASP cells more directly conveyed, especially when it comes to the SASP-cell/T-cell axis. We have now added "we aimed to summarize functional cell types in order to highlight the functional importance of SASP cells and their communicational patterns. Subsequently, we combined [...]" to the Methods (page 21) to point that out more clearly.

Reviewer only. Figure 2. Full depiction of cell types and their interaction strength in CellChat. The full cellular interaction with each cell type demonstrates the prominent interaction of SASP cells with CD8⁺ effector T cells and CD8⁺ naïve T cells.

2. For Fig. 4E & Fig.5D, there is a lack of fair comparison and evaluation of the gene expression about members of MIF and PECAM1 signaling pathway among SASP cells and other cell types.

Response: We agree with the reviewer and are thankful for the opportunity to show the complete MIF and PECAM1 signaling pathways. We added these two to the Fig. 4 associated Supplementary Figure 4 as A and B:

Supplementary Figure 4. *MIF* and *PECAM* pathways in human hematopoietic bone marrow cell types. (a) The *MIF* pathway and its key members show a highly heterogeneous expression pattern among all cell clusters. While $CD10^+$ B cells show a high expression of *MIF*, *CD74* and *CXCR4*, the expression of *CD44* is low. An overall high expression of all *MIF* members is evident in $CD8^+$ effector T cells and conventional dendritic cells and SASP cells. (b) The *PECAM* pathway shows an expression of *PECAM1* in $CD16^+$ monocytes, plasma cells and SASP cells.

3. “The authors performed pseudotime analysis to demonstrate that the SASP cluster, in both bone marrow hematopoietic cells and bone/bone marrow mesenchymal cells, was in the final phases of cellular differentiation. Although, in Fig. 5E, authors coincidence the result with prior knowledge (the increased expression of *Cdkn1a/p21Cip1* and *Trp53*), monocle cannot be used to determine the base state of the trajectory. Authors should use an unbiased trajectory method to confirm this.”

Response: We thank the reviewer for this reasoned thought. Indeed, we used monocle to determine the trajectory state, giving us just two options of trajectory interference (rev=TRUE or rev=FALSE). By taking into account *Cdkn1a* and *Trp53* we tried to reduce this bias as much as possible, and since the original raw fastqs were not provided by the authors, we did not make use of other (unbiased) methods of trajectory interference.

An unbiased way for cellular dynamic analysis, as beautifully demonstrated by La Manno et al. is RNA velocity, which needs raw fastqs to recreate spliced and unspliced mRNA matrices³. Since the bam files were provided, we were able to recreate the fastqs for our single cell dataset from there, using the cellranger pipeline. After that, we made use of bustools and velocity to create the spliced and unspliced matrices of each of the eight samples, finally merging them and

calculating velocity to receive an unbiased trajectory interference. The results are demonstrated in the new Supplementary Fig. S9a-b. As predicted, we found that SASP cells are mostly derived from OLC1, OLC2 and *Lepr*⁺ MSCs (UMAP on the top). When focusing on these four cell types (UMAP on the bottom), it was seen mostly consistent that SASP cells were at the developmental end, originating from *Lepr*⁺ MSCs and OLC1 (upper-left continent) and *Lepr*⁺ MSCs as OLC2 (bottom-right continent). These results confirm our trajectory interference with monocle.

Supplementary Figure 9. *Trajectory interference using velocity (La Manno et al. 2018).* (a) The overall trajectory interference shows that SASP cells are mostly developing from OLC1, OLC2 and *Lepr*⁺ MSCs. (b) A focus on these four cell types reveals the OLC1 and *Lepr*⁺ MSCs as main origin of the upper-left continent, and the OLC2 and *Lepr*⁺ MSCs as the origin of a different SASP cell population in the bottom-right continent.

4. The authors validated the changes of *Mif* mRNA levels in the bone of mouse model. However, the authors also provided an interesting conclusion showing that the *MIF* pathway has been important in both hematopoietic and mesenchymal cells to participate in cellular interactions. Authors should provide validation of this interesting interaction analysis.

Response: We agree with the Reviewer that MIF signaling may be of particular interest in the context of cellular senescence. In this paper, we primarily used this pathway to validate our *in silico* predictions using experimental genetic clearance of senescent cells. Further dissection of this pathway and its possible role in immune evasion by senescent cells is, however, beyond the

scope of the present work and is the subject of future studies in our laboratory. We do address this issue in the Discussion on page 16 where we note that further studies are needed to address this issue.

Minor comments:

1. In methods - Single-cell RNA-seq (scRNA-seq) analysis and page 29, there exists a minor error. “For Fig. 2C,” and “For Fig. 3C” should be “for Fig. 4C,” and “For Fig. 5C”, respectively.

Response: We thank the reviewer for pointing this out and corrected the two mistakes.

2. For Fig. S2 B-C & Fig. S3E, the color legend needs to show the cell-type names, to avoid confusion about the interpretation of the data.

Response: We thank the reviewer for pointing out the missing clarity. To verify that the analyzed cells on the right side in B and in C are SASP cells, we color-coded the font, consistent with the previous figures, and surrounded the whole figure C with a rectangle using the same color (previously Fig. S2B-C, now Supplementary Fig. 3e-f).

Supplementary Fig. 3e-f

For Supplementary Fig. 5e (previously S3e), this color-coding functionality was not provided by Cellchat, so we colored the cluster manually and hope that this way, the cell types are distinct at first glance:

Supplementary Fig. 5e

For consistency, we corrected Supplementary Fig. 7c-d and g in the same manner.

Reviewer 3

This manuscript by Dominik Saul et al. generated a novel gene set (Sen Mayo) including previously reported genes that were enriched in senescent and/or SASP-secreting cells. For RNA-seq, transcriptome-wide gene expression data was obtained from 2 different cohorts and further employed these data to identify the senescent cell in vivo. Meanwhile, the author also validate their hypothesis in single-cell resolution, and combined with trajectory inference methods to explore senescence-associated pathways. Overall, this study provided comprehensive correlative data in a clearly written way, however, there are several concerns addressed as follows:

1. *To determine whether senescence-and SASP-associated pathways were enriched with aging, the authors selected to analyze transcriptional regulatory relationships (Figure 1B). It would be very interesting, if the author can discuss what happens if these transcription factors and regulate signaling molecules are perturbed? This might straightforwardly leverage some public databases and tools, including PMID: 32051003, 28991892, and 25058159. Meanwhile, please provide further explanation regarding how SenMayo encodes a dense network of different protein classes in a strong interaction network (Figure 1F). Tables of cluster analysis results can be additionally included as part of the supplementary section.*

Response: We thank the reviewer for these challenging yet valuable suggestions. The regulatory elements of the SenMayo in total are indeed of major interest and we decided to add both of the reviewer's suggestions, iRegulon and SCENIC, combine them, and leverage these tools for our datasets. This led to the new Supplementary Fig. 1 and 8. Indeed, the major motif controlling the SASP members was Factorbook-NFKB1 (Supplementary Fig. 1), which is in accordance with our Fig. 1B (RELA and NFkB1 are both both NF-κB subunits). Further exploring the associated transcription factors (TF) with iRegulon, BCL3, RXRA and NFIC were the highest enriched (Supplementary Fig. 1b), with the transcription co-activator BCL3 furthermore controlling BCL3, NFKB1, NFKB2, RELA and IKZF1 (Supplementary Fig. 1c). The three main TFs (BCL3, RXRA and NFIC) controlled a major proportion of the SenMayo genes (Supplementary Fig. 1d). To verify

these regulators as substantial, we used SCENIC to first create a tSNE visualization along 50 PCs and 50 perplexity according to the AUCell determined major regulons (Supplementary Fig. 8a). Interestingly, most of the SASP cells can be found within the upper central continent (Fig. Supplementary Fig. 8b).

An overlay with two of the key regulons (BCL3 and RXRA, a third one would get too confusing with the two red colors just distinguishable) upon the new tSNE revealed the iRegulon-predicted importance of both BCL3 and RXRA for the SASP cells (Fig. Fig. S5F).

We thank the reviewer for these very helpful suggestions and have now added “The key regulatory elements of the SenMayo genes according to iRegulon⁴ feature the Factorbook-NFKB1 motif (Suppl. Fig. S5A), and BCL3 (Suppl. Fig. S5B-C) represents the leading transcription factor for a majority of SASP genes (Suppl. Fig. S5D).” and “Interestingly, and as predicted from the human RNA-seq data (Fig. 1B), the SASP cells were mainly controlled by the transcription factor BCL3 (and RXRA, Suppl. Fig. S5E-F).” to the main text (page 6).

In addition, we report the characteristics of the two networks (Fig. 1E-F) and Suppl. Fig. 1 like average number of neighbors, characteristic path length, network heterogeneity and network centralization in the new Suppl. Table. 1.

Supplementary Figure 1. *iRegulon* (Janky et al. 2014) predicts the key regulons for the SASP cells. (a) The motif with the highest enrichment for the SASP genes, according to iRegulon, was Factorbook-NFKB1. (b) The most important transcription factors with 92, 28 and 34 targets within SenMayo were BCL3, RXRA and NFIC, respectively. The first transcription factor controlled (c) BCL3, NFKB1 and -2, RELA and IKZF1, thus confirming the RNA-Seq predictions in the young and old dataset (Fig. 1B). (d) The three transcription factors control a majority (95/125) of SenMayo genes.

Supplementary Figure 8. SCENIC (Aibar et al. 2017) predicts the key regulons for the SASP cells within the human single cell dataset. (a) A regulon-based tSNE is constructed, where the SASP cells contribute substantially to the upper-middle continent (purple). (b) The predicted regulons, BCL3 and RXRA, indeed control the SASP cells containing continent.

2. Since the senolytic drug combination, Dasatinib plus Quercetin (D+Q) in the context of diabetic kidney disease was used to validate the ability of SenMayo on predicting senescent cell clearance (Figure 3), please describe the concept for choosing diabetic kidney disease model to validate the data.

Response: As now noted on page 18, the original trial examined patients with diabetic kidney disease because both obesity (associated with type 2 diabetes mellitus) and chronic kidney disease are linked to an increase in senescent cell burden.

3. In the previous analysis, principal component analysis (PCA) was performed to estimate/evaluate the variance displayed in Figure 3C. What about UMAP variance estimation/evaluation? Further explanations are required to clarify the theories (Figure 4A).

Response: We thank the reviewer for this suggestion. We thought that the genes of interest would be best described in the easiest graphical depiction possible for bulk RNA-seq data, especially since we just looked at the dimension “treatment” with three characteristics (young vs old_veh vs old_ap). But indeed, a umap-representation is used afterwards, so we performed a normalization of the DESeq data, followed by a transformation needed for the umap-package (0.2.7.0) and plotted the treatment groups as UMAP, unfortunately not resulting in a better visual representation compared to the PCA-plot, but still a good distinction of young vs. both old groups (Reviewer only. Fig. 3):

Rev. only. Figure 3. *UMAP depiction of mouse INK-ATTAC RNA-Seq dataset from Fig. 3C.* While the young group can be distinguished from the old_veh and the old_ap group, a distinction between ap-treatment and vehicle-treatment remains more difficult than in the PCA-plot in Fig. 3C.

To continue these thoughts and give more insights into the UMAP representation of the human single cell RNA-seq data from Fig. 4A, we aimed to combine the more gene-focused visual representation of a PCA-plot, with the UMAP from Fig. 4A, leading to a Similarity Weighted Nonnegative Embedding (SWNE) representation of the (before SenMayo-enriched) representation. The leading genes for this SWNE plot are the SenMayo genes:

Reviewer only Figure 4. (A) SWNE depiction of the UMAP from Fig. 4A. All SenMayo genes are embedded within the SWNE and represent the places with the “highest” expression of a specific gene. (B) Examples of expression of CD9 (right part, coloured in red) and (C) CCL4 (bottom-left part, coloured in red) in the SNWE representation.

These analyses give an interesting idea on the importance of different genes for the individual clusters. However, we feel that for the reader, this would be more confusing than the original Fig. 4A, in which the general enrichment is shown. Also, a strength of the SenMayo is that all SASP genes are enriched simultaneously, and not one single gene is of higher importance than others.

4. How the cell types from single-cell transcriptomic data presented in Figures 4A and 5A were identified? Please explain in detail the methods for cell-type identification from single-cell transcriptomic data. Similar interpretation of scRNA-seq cell type annotation protocol presented in figures 4C and 5C. Since cell type annotation obtained from a previous study (PMID: 30518681) which is lack of canonical genes references for each cluster, should the new cluster referred as “SASP cells” generated by selecting top 10% highest expression of senescence/SASP-associated genes technically be aligned together in the whole UMAP representation? Besides that, please describe in more detail regarding the interaction values between SASP and the other particular cell types. The two aging signatures CellAge and GenAge and its significant correlation to SASP also need to be interpreted more clearly.

Response: The clustering for the human dataset ⁵ was obtained as follows, and in now added to the methods (page 21): “After quality control and integration, we performed a modularity optimized Louvain clustering with the resolution “1.4”, leading to 40 distinct clusters in the human dataset. Subsequently, we performed the labelling for these 40 clusters manually with established key marker genes (Suppl. Fig. 3a).” We also added to the methods section:

“For the murine dataset, we chose the same order of analysis, and picked the resolution “1.4”, leading to 40 distinct clusters, which were manually assigned according to established marker genes (Suppl. Fig. 6a).”

However, as the reviewer pointed out, there were no canonical reference genes in this paper. In order to enable the reader to reproduce these results, we added a dotplot highlighting the canonical genes for each cluster, along with the underlying references:

Reviewer only. Figure 5. Dotplot depicting the canonical genes per cluster. They key genes per cluster are demonstrated in both the (A) human dataset along with the reference for each marker and cell type and in the (B) murine dataset along with references for each marker and cell type.

We have added the human dotplot as Suppl. Fig. 3a, and the murine dotplot as Suppl. Fig. 6a. To assure that the reader can reproduce the clustering for both the human and murine dataset, we also added a notebook for the Seurat-file acquisition until the clustering (“notebook_human_murine_seurat_acquisition.Rmd”, accessible via <https://datadryad.org/stash/share/YdD6C2ZFDgSizXehPR0qqPy4io7oRQJMGRIrPuij9WU>).

Since the SASP cells are generated by enriching for SenMayo genes, we do not necessarily expect the same cell type to enrich for these genes. Subsequently, some clusters with the capability of secreting a SASP phenotype, like monocytic cells would be more expected to enrich in SenMayo, while others like erythroid progenitors would be less expected to enrich for SenMayo. Thus, the highest enrichment in certain parts of the continents is expected, but an occasionally occurring enrichment in others is also not unexpected.

For the interaction values in Fig. 4C and Fig. 5C, we added “the numbers represent the relative interaction strength as sum of interaction weights. Edge weights are proportional to interaction strength, and a thicker line refers to a stronger signal²”.

We agree that the interaction between genAge and SenMayo as CellAge and SenMayo with plotted three variables may not be as clear as intended at first sight. We subsequently chose to plot them separately with a spearman correlation: For SenMayo and genAge, we found a significant correlation with an R of 0.43, while for SenMayo and CellAge, this was R=0.35 in the human dataset. In the murine dataset, the R was 0.61 for SenMayo and genAge and R=0.67 for SenMayo and CellAge, respectively (Reviewer only. Fig. 6, and added to Supplementary Fig. 3c,d and Supplementary Fig. 6b,c).

Reviewer only. Figure 6. Bivariate correlation plots for the human (A, B) and murine (C, D) dataset. Both SenMayo and genAge show reliable correlations (A: R=0.43, C: R=0.61) as do SenMayo and CellAge (B: R=0.35, D: R=0.67).

5. Please present the reconstruction and basis of pseudo-time analysis in a more detailed manner, along with the procedure/protocol of trajectory analysis on the monocle package. The three main origins for the SASP cluster, namely *Lepr+* MSCs, OLC 1, and OLC 2, were depicted in pseudo time by applying trajectory inference. Why only canonical markers of senescence were reported in Figure 5B other than the result of *TGFB1* in a terminal developmental branch?

Response:

Please see Rev. 2 – point 3

We thank the reviewer for this reasoned thought. Indeed, we used monocle to determine the trajectory state, giving us just two options of trajectory interference (rev=TRUE or rev=FALSE). By taking into account *Cdkn1a* and *Trp53* we tried to reduce this bias as much as possible, and since the original raw fastqs were not provided by the authors, we did not make use of other (unbiased) methods of trajectory interference.

An unbiased way for cellular dynamic analysis, as beautifully demonstrated by La Manno et al. is RNA velocity, which needs raw fastqs to recreate spliced and unspliced mRNA matrices³. Since the bam files were provided, we were able to recreate the fastqs for our single cell dataset from there, using the cellranger pipeline. After that, we made use of bustools and velocity to create the spliced and unspliced matrices of every of the eight samples, finally merging them and

calculating velocity to receive an unbiased trajectory interference. The results are demonstrated in the new Supplementary Fig. S7A-B. As predicted, we found that SASP cells are mostly derived from OLC1, OLC2 and Lepr⁺ MSCs (UMAP on the top). When focusing on these four cell types (UMAP on the bottom), it was seen mostly consistent that SASP cells were at the developmental end, originating from Lepr⁺ MSCs and OLC1 (upper-left continent) and Lepr⁺ MSCs as OLC2 (bottom-right continent). These results confirm our trajectory interference with monocle.

Supplementary Figure 9. *Trajectory interference using velocity (La Manno et al. 2018).* (a) The overall trajectory interference shows that SASP cells are mostly developing from OLC1, OLC2 and Lepr⁺ MSCs. (b) A focus on these four cell types reveals the OLC1 and Lepr⁺ MSCs as main origin of the upper-left continent, and the OLC2 and Lepr⁺ MSCs as the origin of a different SASP cell population in the bottom-right continent.

We provide the underlying R code for the velocity calculations in the notebook "R_notebook_Fig4_5_sup2to10.Rmd" (accessible via <https://datadryad.org/stash/share/YdD6C2ZFDgSizXehPR0qqPy4io7oRQJMGRlrPuij9WU>).

Since the expression of Tgfb1 along pseudotime was explicitly requested, we plot it here (Reviewer only. Fig. 8):

Reviewer only. Figure 8. *Tgfb1* expression along pseudotime. The inlay on the top-left shows the general pseudotime development (originating in the bottom-right and ending in both bottom-left and middle-top). The *Tgfb1* expression is enriched in the end of the pseudotemporal development.

In addition, we provide the R code for the pseudotime calculations in the notebook “R_notebook_Fig4_5_sup2to10.Rmd” accessible via <https://datadryad.org/stash/share/YdD6C2ZFDgSizXehPR0qqPy4io7oRQJMGRlrPuij9WU>.

6. To further test SenMayo in single-cell datasets and potentially contrast bone marrow hematopoietic cells to bone/bone marrow mesenchymal cells, a published murine dataset containing scRNA-seq data from bone and bone marrow mesenchymal cells was leveraged. Why the authors did not select the human bone marrow mesenchymal cells related dataset to avoid biological heterogeneity? Regarding the selection of canonical SASP markers or SASP/senescence markers, verification of existing references may also improve the reproducibility of the current study.

Response: We thank the reviewer for the opportunity to clarify this. Indeed, we think that the applicability of SenMayo in both human and murine datasets is a strength of SenMayo. In our own mouse studies, we frequently faced the obstacle of not having a reliable, yet multifaceted tool to reveal cells with a SASP-like transcriptome in RNA- and single cell RNA-seq datasets. We aimed to establish a panel that can be used in bulk RNA-seq and single cell RNA-seq datasets to uncover SASP-like cells and monitor a treatment effect in a reliable manner.

To further verify an existing reference to improve the reproducibility of the current study within the bone marrow, we leveraged the Tabula Muris Senis single cell dataset⁶ and combined one month and three old month mice as “young” and 24 month and 30 month old mice as “old”. We saw an increase of SenMayo in the old mice’s bone marrow (****, Reviewer only. Fig. 9).

Reviewer only. Figure 9. Comparison of bone marrow from young (1+3m) and old (24 and 30m) mice from the *tabula muris senis*⁶. Enriching for SenMayo, the old mice showed an overall higher enrichment score compared to the young mice (Wilcoxon rank-sum test, $p < 0.001$).

7. The author used Seurat algorithm to analyze 500 unique molecular identifiers (UMIs) and data were obtained based on the following thresholds: \log_{10} genes per UMI > 0.8 , > 250 genes per cell, a mitochondrial ratio of less than 20%. Since results from Sequencing Data Alignment often undergo considerable fluctuations due to modification in specific parameters and tuning. Therefore, without detailed description of how the algorithms were applied, it may seem very difficult to reproduce these results. For example, how did the algorithms result in the number of subgroups and objects in each analysis and how was it concluded that these subgroups are statistically robust? To avoid being cluttered with unnecessary details, the authors may provide a summarized list in either the Supplementary or Code Availability section.

Response: We thank the reviewer for these helpful suggestions. For both the human and murine single cell datasets, we now provide a notebook with which both of these Seurat objects can be acquired and generated the way we did for the manuscript ("notebook_human__murine_seurat_acquisition.nb.html"). Likewise, we provide a notebook for the RNA-Seq (GSE72815_YOE_Notebook) and for the creation of Fig. 4-5 as well as Suppl. 2-10 (R_notebook_Fig4_5_sup2to10, accessible via <https://datadryad.org/stash/share/YdD6C2ZFDgSizXehPR0qqPy4io7oRQJMGRlrPuij9WU>).

Minor comments:

1. All images are presented in a high-resolution and professional manner, however, to be more intelligible without reference, figure legends should provide sufficient details (such as the meaning of including solid line, dashed line, dash-dotted line, dotted line...) corresponding to each component part.

Response: We thank the reviewer for the opportunity to clarify and improve our figure legends. Specifically, we added "arrows point the direction of these interactions" to Fig. 1E and "grey lines represent interactions" to Fig. 1F and "The highlighted genes represent variables, and the arrows

drawn from the origin indicate their “weight” in different directions, according to the theories of Gabriel (<https://doi.org/10.2307/2334381>)” to Fig. 3C, as “the numbers represent the relative interaction strength as sum of interaction weights. Edge weights are proportional to interaction strength, and a thicker line refers to a stronger signal²” to Fig. 4C, and “the numbers represent the relative interaction strength as sum of interaction weights. Edge weights are proportional to interaction strength, and a thicker line refers to a stronger signal²” to Fig. 5C and “the solid line represents the expression values as a function of pseudotime-progress” to Fig. 5E.

2. In Figure S4D, it is unclear how GO/KEGG pathway enrichment is analysed. Please also make sure the color codes clarify in these figures and tables. Meanwhile, this would make sense to state as an advantage if the study had indeed identified a novel pathway or any interaction not previously known, either via bulk or single-cell seq data.

Response: We added “The gene ontology and KEGG analyses (Supplementary Fig. 3f, Supplementary Fig. 7d) with gprofiler2 were done for homo sapiens (hsapiens) and mus musculus (mmusculus) after the positively regulated genes were selected with the “FindMarkers” function (used test=“MAST”, logfc.threshold = 0.25). For the gost function, we used an user threshold of 0.05 and fdr correction method.” to the methods section to clarify the GO/KEGG analyses.

We color-coded the Fig S4C, D and G (now Supplementary Fig. 3e,f and Supplementary Fig. 7c-d) with the color for the SASP cells to be consistent and verify which cells were analyzed. Since we carved out a new cellular component, “SASP cells” from single cell data, their interactions with other cell types (as expected from cells that secrete potentially detrimental cytokines) are novel. However, we regret that in neither bulk nor single cell seq-data, we discovered new pathways. Nonetheless, we analyzed which genes – apart from the already selected 125 SenMayo genes – would be upregulated in our SASP cell cluster (Reviewer only. Fig. 10).

Reviewer only. Figure 10. Top genes in the SASP cells cluster, that were not included in the SenMayo, sorted along their fold average-fold change compared to all other clusters.

The highest fold-change was detected for the genes S100A9, CST3 and TYROBP. To further characterize the importance of these genes, and characterize the SASP cells in more detail, as elucidate their secretory phenotype in more detail, is part of ongoing studies in our lab.

We are submitting our extensively revised manuscript for further consideration and hope it is now acceptable for publication in *Nature Communications*.

Sincerely,

Dominik Saul
Sundeep Khosla

References

1. Lopes-Paciencia, S. *et al.* The senescence-associated secretory phenotype and its regulation. *Cytokine* **117**, 15–22; 10.1016/j.cyto.2019.01.013 (2019).
2. Jin, S. *et al.* Inference and analysis of cell-cell communication using CellChat. *Nature communications* **12**, 1088; 10.1038/s41467-021-21246-9 (2021).
3. La Manno, G. *et al.* RNA velocity of single cells. *Nature* **560**, 494–498; 10.1038/s41586-018-0414-6 (2018).
4. Janky, R. *et al.* iRegulon: from a gene list to a gene regulatory network using large motif and track collections. *PLoS computational biology* **10**, e1003731; 10.1371/journal.pcbi.1003731 (2014).
5. Oetjen, K. A. *et al.* Human bone marrow assessment by single-cell RNA sequencing, mass cytometry, and flow cytometry. *JCI insight* **3**; 10.1172/jci.insight.124928 (2018).
6. The Tabula Muris Consortium. A single-cell transcriptomic atlas characterizes ageing tissues in the mouse. *Nature* **583**, 590–595; 10.1038/s41586-020-2496-1 (2020).

Reviewers' Comments:

Reviewer #1:

Remarks to the Author:

Thank you for revising. I find that you have responded satisfactorily to my comments.

Reviewer #2:

Remarks to the Author:

The authors have suitably addressed my concerns. I agree that it can set a new reference on identifying senescent cells.

Reviewer #3:

Remarks to the Author:

The authors addressed most of the reviewers' concerns; now they are more readable and make sense. Especially the addition of the iRegulon/ SCENIC and data generated with this single-cell technology is a great improvement of the manuscript. Meanwhile, the author also provided the R code in the notebook for further development to enhance throughput, therefore, I recommend acceptance of the manuscript for publication.